# Upstream open reading frames buffer translational variability during *Drosophila* evolution and development

**Yuanqiang Sun**[1†], **Yuange Duan**[1†], **Peixiang Gao**[1], **Chenlu Liu**[1], **Kaichun Jin**[1], **Shengqian Dou**[2], **Wenxiong Tang**[1], **Hong Zhang**[3], **Jian Lu**[1,4,5]*

[1]State Key Laboratory of Gene Function and Modulation Research, Center for Bioinformatics, School of Life Sciences, Peking University, Beijing, China; [2]Eye Institute of Shandong First Medical University, State Key Laboratory Cultivation Base, Shandong Provincial Key Laboratory of Ophthalmology, Beijing, China; [3]College of Ecology, Lanzhou University, Lanzhou, China; [4]Beijing Advanced Center of RNA Biology (BEACON), Peking University, Beijing, China; [5]Southwest United Graduate School, Kunming, China

**\*For correspondence:**
luj@pku.edu.cn

[†]These authors contributed equally to this work

**Competing interest:** The authors declare that no competing interests exist.

## eLife Assessment

This study reveals the **important** role of upstream open reading frames (uORFs) in limiting the translational variability of downstream coding sequences. Through a combination of computational simulations, comparative analyses of translation efficiency across different developmental stages in two closely related *Drosophila* species, and manipulative, experimental validation of translation buffering by an uORF for a gene, the authors provide **convincing** evidence supporting their conclusions. This work will be of broad interest to molecular biologists and geneticists.

**Abstract** Protein abundance tends to be more evolutionarily conserved than mRNA levels both within and between species, yet the mechanisms underlying this phenomenon remain largely unknown. Upstream open reading frames (uORFs) are widespread *cis*-regulatory elements in eukaryotic genomes that regulate translation, but it remains unclear whether and how uORFs contribute to stabilizing protein levels. In this study, we performed ribosome translation simulations on mRNA to quantitatively assess the extent to which uORF translation influences the translational variability of downstream coding sequences (CDSs) across varying contexts. Our simulations revealed that uORF translation dampens CDS translational variability, with buffering capacity increasing in proportion to uORF translation efficiency, length, and number. We then compared the translatomes at different developmental stages of two *Drosophila* species, demonstrating that uORFs buffer mRNA translation fluctuations during both evolution and development. Experimentally, deleting a uORF in the *bicoid* (*bcd*) gene—a prominent example of translational buffering—resulted in extensive changes in gene expression and phenotypes in *Drosophila melanogaster*. Additionally, we observed uORF-mediated buffering between primates and within human populations. Together, our results reveal a novel regulatory mechanism by which uORFs stabilize gene translation during development and across evolutionary time.

## Introduction

Organisms have evolved various strategies for the spatiotemporal regulation of gene expression (*Buccitelli and Selbach, 2020*; *Lee and Young, 2013*; *Macneil and Walhout, 2011*). This is important because aberrant gene expression can result in phenotypic defects or diseases, while the variation and evolution of gene expression patterns frequently promote phenotypic diversification and adaptation (*Hill et al., 2021*). Although variations in mRNA abundance are widely observed within or between species, protein abundance tends to show stronger evolutionary constraint (*Signor and Nuzhdin, 2018*; *Vogel, 2013*), as observed in yeasts (*Artieri and Fraser, 2014*; *McManus et al., 2014*; *Wang et al., 2015*), primates (*Khan et al., 2013*; *Wang et al., 2018*), and other organisms (*Kusnadi et al., 2022*; *Laurent et al., 2010*; *Schrimpf et al., 2009*). Nevertheless, the molecular mechanisms by which the conservation of protein abundance across species is achieved are largely unknown (*Signor and Nuzhdin, 2018*; *Vogel, 2013*).

Eukaryotic mRNA translation is a crucial step in gene expression and is highly regulated by multilayered mechanisms (*Jackson et al., 2010*; *Sonenberg and Hinnebusch, 2009*; *Teixeira and Lehmann, 2019*). Upstream open reading frames (uORFs), which are short open reading frames in the 5-terminal untranslated regions (5′ UTRs) of eukaryotic mRNAs, play crucial roles in regulating mRNA translation. Approximately 50% of eukaryotic genes contain uORFs (*Zhang et al., 2021*), and their evolution has been tightly shaped by natural selection (*Churbanov et al., 2005*; *Neafsey and Galagan, 2007*; *Resch et al., 2009*; *Zhang et al., 2018a*; *Zhang et al., 2019*). The functions of uORFs have been explored in various contexts, including development (*Chen et al., 2017*; *Cheng et al., 2018*; *Komonyi et al., 2005*; *Kurihara et al., 2018*; *Malzer et al., 2013*; *Medenbach et al., 2011*; *Yang et al., 2017*; *Zhang et al., 2018a*), disease (*Calvo et al., 2009*; *Lee et al., 2021*; *Liu et al., 1999*; *Wen et al., 2009*; *Wiestner et al., 1998*), and stress responses (*Andreev et al., 2018*; *Andreev et al., 2015*; *Bohlen et al., 2020*; *Costa-Mattioli and Walter, 2020*; *Hinnebusch, 2005*; *Lu et al., 2004*; *Vasudevan et al., 2020*; *Vattem and Wek, 2004*). The prevailing consensus is that uORFs typically repress downstream coding sequence (CDS) translation by sequestering ribosomes, a process influenced by factors such as uORF length, position, and sequence context (*Andreev et al., 2018*; *Calvo et al., 2009*; *Hinnebusch et al., 2016*; *Johnstone et al., 2016*; *Kozak, 1989*; *Morris and Geballe, 2000*; *Young and Wek, 2016*). However, under stress conditions, certain uORFs can facilitate CDS translation by promoting ribosome reinitiation, illustrating their context-dependent functions (*Andreev et al., 2018*; *Andreev et al., 2015*; *Baird et al., 2014*; *Chen et al., 2010*; *Dever et al., 1992*; *Palam et al., 2011*; *Young and Wek, 2016*).

Gene expression noise, which arises from the inherent stochasticity of biological processes such as transcription and translation, is generally detrimental to organismal fitness (*Fraser et al., 2004*) and is primarily determined at the translational level (*Thattai and van Oudenaarden, 2001*). Recent studies suggest uORFs might play essential roles in buffering translational noise and stabilizing protein expression. For example, *Wu et al., 2022* demonstrated that uORFs reduce protein production rates to stabilize TOC1 protein levels, ensuring precise circadian clock function in plants (*Wu et al., 2022*). Similarly, *Bottorff et al., 2022* used a human cell reporter system to show that a ribosome stall in cytomegaloviral *UL4* uORFs buffers against CDS translation reductions (*Bottorff et al., 2022*). Under stress, translation initiation is typically downregulated, yet most human mRNAs resistant to this inhibition contain translated uORFs, with a single uORF often being sufficient for resistance (*Andreev et al., 2015*). The computational model of Initiation Complexes Interference with Elongating Ribosomes (ICIER) suggests that derepression of downstream translation is a general mechanism of uORF-mediated stress resistance (*Andreev et al., 2018*). Despite these findings, the current understanding of uORFs in stabilizing translation is limited to single-gene cases or stressed conditions. It remains unclear whether and how uORFs affect gene translation variability on a genome-wide scale during evolution and development, and whether the identified mechanisms are universal or vary substantially among different taxa. To address these questions, a combination of modeling, genome-wide analyses, and comparative studies across species is required.

In this study, we first adapted the ICIER framework (*Andreev et al., 2018*) to simulate the translating ribosome on an mRNA to quantitatively measure the extent to which uORF translation reduces the translational variability of the downstream CDS under different translation contexts. We then compared the translatomes of two closely related *Drosophila* species, *D. melanogaster* and *D. simulans*, and further supported the notion that uORFs could buffer the fluctuations of CDS translation

during the development and evolution of *Drosophila*. The patterns also reappeared among primates and human populations. We next knocked out the *bicoid* (*bcd*) uORF, a case showing significant buffering effect in our data, and observed broad changes in the embryonic transcriptome and phenotypic defects in *D. melanogaster*. Together, our results demonstrate a novel role for uORFs in maintaining translation stability during *Drosophila* evolution and development.

## Results

### An extended ICIER model for quantifying uORF buffering in CDS translation

To quantitatively assess how uORF translation modulates the variability of downstream CDS translation, we adapted the ICIER model (*Andreev et al., 2018*), originally grounded in the totally asymmetric simple exclusion process (TASEP). TASEP has been extensively utilized to model the stochastic nature of ribosome movement along mRNA, capturing the effects of ribosome traffic jams, where ribosomes may slow down or stall when a site ahead is occupied (*Ciandrini et al., 2010*; *Reuveni et al., 2011*; *von der Haar, 2012*; *Zhao and Krishnan, 2014*). The ICIER model extends this by simulating the interplay between scanning (40S) and elongating (80S) ribosomes, particularly focusing on how uORFs impact the overall translation rate of the main CDS (*Andreev et al., 2018*).

We extended the original ICIER model (*Andreev et al., 2018*) with several major modifications (*Figure 1A*). First, while the original ICIER model only considered the scenario where the elongating ribosome (80S) causes downstream scanning ribosomes (40S) to dissociate from the mRNA when they move along the mRNA and collide, recent findings have shown that upstream dissociation can also play a critical role in uORF-mediated regulation (*Bottorff et al., 2022*). To incorporate this, we accounted for more complex ribosome interactions, including three possible scenarios where the 80S collides with 40S: (i) 80S only causes the downstream 40S to dissociate from the mRNA with a probability of $K_{down}$ ('downstream dissociation', $K_{down}$ ranging from 0 to 1), following the original ICIER model; (ii) the 80S only causes the upstream 40S to dissociate from the mRNA with a probability of $K_{up}$ ('upstream dissociation', $K_{up} > 0$ and $K_{down} = 0$); and (iii) a combination of the downstream and upstream dissociation models ('double dissociation', $K_{up} > 0$ and $K_{down} > 0$).

Second, the original ICIER model only considered the ribosome collision and dissociation in the uORF and counted the 40S scanning ribosome escaping from the uORF as a proxy of the CDS translation rate. In our extended model, we also considered the ribosome collision and dissociation events in the CDS downstream of the uORF. We recorded the number of 80S ribosomes that completed translation at the stop codon of a CDS ($N_{EC}$) or uORF ($N_{Eu}$) during a given time interval, using these counts as proxies to quantify the translation rate of CDS or uORF. These indices allowed us to directly and quantitatively measure the impact of uORFs on CDS translation.

Third, while the original ICIER model accounted for only a single uORF within an mRNA, we extended it to consider two uORFs coexisting on an mRNA molecule, allowing us to explore the possible combinatorial effects of uORF-mediated translation regulation in a more complex yet more common scenario, as previous studies have shown that uORFs tend to be clustered within genes (*Zhang et al., 2021*).

These extensions allow for a more comprehensive exploration of uORF-mediated translational buffering, offering deeper insights into how these regulatory elements might stabilize protein synthesis across varying translation contexts.

### uORF-mediated buffering of CDS translation across different parameter settings

To systematically investigate the extent to which uORF translation modulates the variability of downstream CDS translation, we conducted simulations across various parameter settings using three dissociation models: upstream, downstream, and double dissociation, each reflecting different possible interactions between scanning and elongating ribosomes on the mRNA. We considered a range of parameters crucial to the translation process (*Supplementary file 1*), including the length of the 5′ leader before the uORF (fixed at 150 nucleotides), the length of the uORF itself (ranging from 2 to 100 codons), the distance between the uORF stop codon and the CDS start codon (150 nucleotides), the length of the CDS (500 codons), and the length of the 3′ UTR (150 nucleotides). Additionally, we

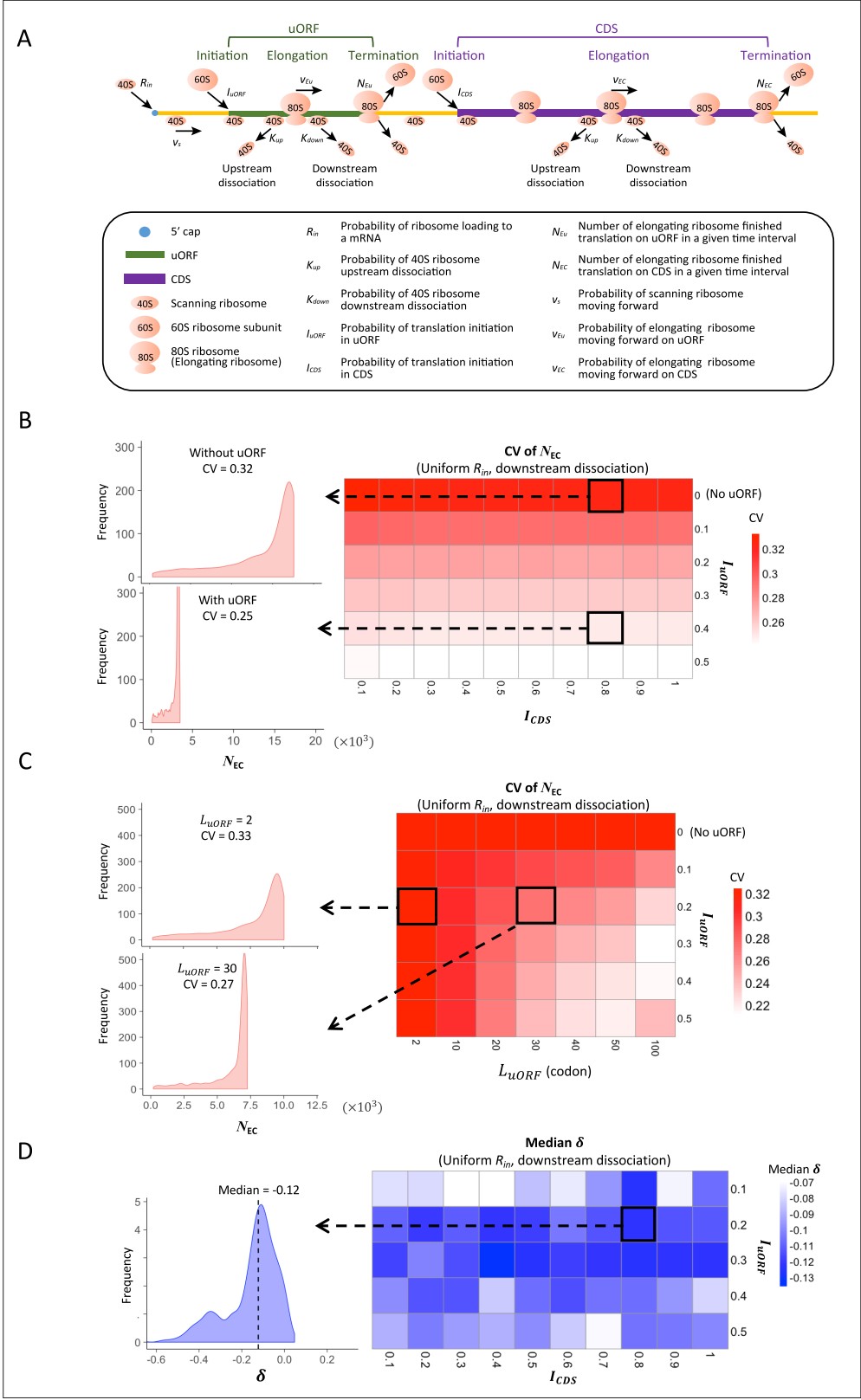

**Figure 1.** Modeling simulation of uORF-mediated translation buffering. (**A**) Model schema of the modified ICIER model (on the top). The parameters are listed in the box below the schema. (**B**) Heatmap showing the CVs of CDS translation rate ($N_{EC}$) under different $I_{CDS}$ (x-axis) and $I_{uORF}$ (y-axis) combinations with a uniform distribution of $R_{in}$ input and the downstream dissociation model. The left panels elicited by the dotted lines from specific

*Figure 1 continued on next page*

*Figure 1 continued*

squares of the right heatmap were two examples showing the distribution of $N_{EC}$ under $I_{CDS} = 0.8$ and $I_{uORF} = 0$ (top panel, without uORF) and $I_{CDS} = 0.8$ and $I_{uORF} = 0.4$ (bottom panel, with uORF). (**C**) Heatmap showing CVs of CDS translation rate ($N_{EC}$) under different $\boldsymbol{L_{uORF}}$ (x-axis) and $I_{uORF}$ (y-axis) combinations with a uniform distribution of $R_{in}$ input and the downstream dissociation model. The left panels elicited by the dotted lines from specific squares of the right heatmap were two examples showing the distribution of $N_{EC}$ under $L_{uORF} = 2$ and $I_{uORF} = 0.2$ (top panel) and $L_{uORF} = 30$ and $I_{uORF} = 0.2$ (bottom panel). (**D**) Heatmap showing median $\delta$ under different $I_{CDS}$ (x-axis) and $I_{uORF}$ (y-axis) combinations with a uniform distribution of $R_{in}$ input and the downstream dissociation model. The left panel elicited by the dotted line from a specific square of the right heatmap was an example showing the distribution of $\delta$ under $I_{CDS} = 0.8$ and $I_{uORF} = 0.2$. The vertical dashed line indicated the median value of $\delta$.

The online version of this article includes the following source data and figure supplement(s) for figure 1:

**Source data 1.** Raw numerical data underlying the figures.

**Figure supplement 1.** The input distribution and model simulation flow.

**Figure supplement 1—source data 1.** Raw numerical data underlying the figures.

**Figure supplement 2.** The median of CDS translation rate under uniform distribution of $R_{in}$.

**Figure supplement 2—source data 1.** Raw numerical data underlying the figures.

**Figure supplement 3.** The median of CDS translation rates under exponential distribution of $R_{in}$.

**Figure supplement 3—source data 1.** Raw numerical data underlying the figures.

**Figure supplement 4.** The CVs of CDS translation rates under uniform distribution of $R_{in}$.

**Figure supplement 4—source data 1.** Raw numerical data underlying the figures.

**Figure supplement 5.** The CVs of CDS translation rate under exponential distribution of $R_{in}$.

**Figure supplement 5—source data 1.** Raw numerical data underlying the figures.

**Figure supplement 6.** The CVs of CDS translation rates across different uORF length under uniform distribution of $R_{in}$.

**Figure supplement 6—source data 1.** Raw numerical data underlying the figures.

**Figure supplement 7.** The CVs of CDS translation rates across different uORF length under uniform distribution of exponential distribution of $R_{in}$.

**Figure supplement 7—source data 1.** Raw numerical data underlying the figures.

**Figure supplement 8.** The correlations between changes of uORF translation rate and downstream CDS translation rate under uniform distribution of $R_{in}$.

**Figure supplement 8—source data 1.** Raw numerical data underlying the figures.

**Figure supplement 9.** The correlations between changes of uORF translation rate and downstream CDS translation rate under exponential distribution of $R_{in}$.

**Figure supplement 9—source data 1.** Raw numerical data underlying the figures.

**Figure supplement 10.** The median $\delta$ under uniform distribution of $R_{in}$.

**Figure supplement 10—source data 1.** Raw numerical data underlying the figures.

**Figure supplement 11.** The median $\delta$ under exponential distribution of $R_{in}$.

**Figure supplement 11—source data 1.** Raw numerical data underlying the figures.

**Figure supplement 12.** Two-uORF model simulation measurements.

**Figure supplement 12—source data 1.** Raw numerical data underlying the figures.

**Figure supplement 13.** Comparison of the buffering effects between single uORF and two uORFs.

**Figure supplement 13—source data 1.** Raw numerical data underlying the figures.

---

modeled the probabilities associated with ribosome movement and initiation, such as the probability of a 40S ribosome moving to the next nucleotide ($\nu_s$=0.3), the probability of an 80S ribosome moving to the next position within the uORF ($\nu_{Eu}$=0.3) and within the CDS ($\nu_{EC}$=0.5), and the probability of loading a new 40S ribosome at the 5' end of the mRNA ($R_{in}$). We also explored different probabilities of translation initiation at both the uORF start codon ($I_{uORF}$) and the CDS start codon ($I_{CDS}$). Key parameters were adapted from the original ICIER model (***Andreev et al., 2018***), ensuring a robust basis for comparison while allowing exploration of additional variables that influence uORF-mediated translational buffering.

In our simulations, $R$ values were varied to simulate fluctuations in translational resources, such as ribosome availability, that could arise from genetic differences or environmental changes during evolution or development. By generating 1000 $R$ values following either uniform or exponential distributions (ranging from 0 to 0.1, *Figure 1—figure supplement 1A*), we aimed to capture the natural variability in ribosome loading rates that might occur across different species, individuals, or developmental stages. These values were then fed into the simulation models to evaluate their impact on both the level and variability of translation rate for CDSs, with the number of ribosomes completing translation on CDSs ($N_{EC}$) in a given time interval used as a proxy for translational rate (*Figure 1—figure supplement 1B*).

Across all model settings, uORF translation ($I_{uORF}>0$) consistently reduced CDS translation rate ($N_{EC}$) by about 30–80% as $I_{uORF}$ increased from 0.1 to 0.5, compared to scenarios where the uORF was absent or untranslated ($I_{uORF}=0$; *Figure 1—figure supplements 2 and 3*). This confirms the inhibitory effect of uORFs on downstream CDS translation. Notably, the coefficient of variation (CV), a measurement of variability for translation rate ($N_{EC}$), was lower when the uORF was translated (*Figure 1B*). These CV values further decreased by approximately 10% to 25% as $I_{uORF}$ increased from 0.1 to 0.5 (*Figure 1B*), and this buffering effect persisted across different parameter settings (*Figure 1—figure supplements 4 and 5*). Moreover, the CV of translation rate ($N_{EC}$) further decreased by about 6–30% as uORF length ($L_{uORF}$) increased from 2 to 100 codons (*Figure 1C*, *Figure 1—figure supplements 6 and 7*). These simulation results indicate that uORF translation can reduce variability in downstream CDS translation, with the buffering capacity positively correlated with both uORF translation initiation efficiency and length under the applied simulation conditions.

To more quantitatively investigate the relationship between changes in translation rate in uORF ($N_{EU}$) and CDS ($N_{EC}$), for each parameter setting, we used the median value ($R_{inm}$) of the 1000 $R_{in}$ inputs as the baseline, calculating the corresponding $N_{ECm}$ and $N_{EUm}$ values. We then calculated the changes of $N_{EC}$ ($\Delta N_{EC}$) and $N_{EU}$ ($\Delta N_{EU}$) relative to $N_{ECm}$ and $N_{EUm}$ for each other $R_{in}$ value, respectively. Across different parameter settings, $\Delta N_{EU}$ was consistently and significantly positively correlated with $\Delta N_{EC}$ (p<0.001, Spearman's correlation; *Figure 1—figure supplements 8 and 9*), indicating that fluctuations of the ribosome loading rate influence the translation of both uORFs and CDSs in the same direction. Nevertheless, our simulations showed that variations in $R_{in}$ led to a larger change in $N_{EU}$ than in $N_{EC}$, as the median value of $\delta$ [defined by $log_2 \left( \Delta N_{EC}/\Delta N_{EU} \right)$] was consistently less than 0 across various $I_{uORF}$ and $I_{CDS}$ combinations (*Figure 1D*, *Figure 1—figure supplements 10 and 11*). This finding suggests that translational fluctuation in uORFs is greater than that in downstream CDSs, indicating that uORFs can buffer against upstream fluctuations.

Simulations of mRNAs with two uORFs revealed patterns consistent with a buffering role for uORFs (*Figure 1—figure supplement 12*). To compare the buffering effects of a single uORF versus two uORFs, we calculated the ratio of the CV of $N_{EC}$ with two uORFs to that with a single uORF. A ratio less than 1 suggests that two uORFs provide greater buffering than a single uORF. For comparability, we examined the CV of $N_{EC}$ where the $I_{uORF}$ in the single-uORF model equals $I_{uORF1}$ ($I_{uORF}$ of the first uORF) in the two-uORF model, both ranging from 0 to 0.5. The CV ratio consistently remained below 1 across a range of $I_{uORF2}$ ($I_{uORF}$ of the second uORF; *Figure 1—figure supplement 13*), indicating that two uORFs offer stronger buffering than a single uORF.

Collectively, these simulations collectively suggest: (1) uORF-mediated translational control buffers against CDS translation variability, (2) uORFs exhibit greater translational fluctuations than downstream CDSs, and (3) the buffering capacity of uORFs positively correlates with their translation initiation efficiency and length. Subsequently, we sought to validate these simulation results in a biological context by confirming the uORF-mediated buffering effect during organismal evolution and development, as both processes face environmental and/or genetic changes that frequently disturb mRNA translation.

## Generating matched translatome data from two *Drosophila* species for comparative analysis

To validate the uORF-mediated translational buffering during *Drosophila* evolution, we performed a comparative analysis of the translatomes of two closely related *Drosophila* species, *D. melanogaster* and *D. simulans*, which diverged approximately 5.4 million years ago (*Tamura et al., 2004*). We generated high-throughput sequencing data for *D. simulans*, including transcriptome (mRNA-Seq) and

translatome (Ribo-Seq) profiles from various developmental stages and tissues, including embryos at 0–2 hr, 2–6 hr, 6–12 hr, and 12–24 hr, third-instar larvae, P7–8 pupae, female and male bodies, and female and male heads. In total, we obtained approximately 786 million high-quality reads for *D. simulans* (*Supplementary file 2*). These datasets were designed to be directly comparable to the previously published *D. melanogaster* data (*Zhang et al., 2018a*), with identical embryonic stages and tissue types. For each sample, we followed established procedures (*Chothani et al., 2022*; *Ingolia, 2016*; *Ingolia et al., 2009*; *Ingolia et al., 2011*; *Zhang et al., 2018a*) to calculate the translational efficiency (TE) for each feature (CDS or uORF). TE serves as a proxy for the translation rate at which ribosomes translate mRNA into proteins, typically quantified by comparing the density of ribosome-protected mRNA fragment (RPF) to the mRNA abundance for a feature (see Materials and methods). This comprehensive comparative translatome analysis, utilizing matched developmental stages and tissues between the two species, allowed us to assess the uORF-mediated translational buffering effects during evolution.

## Translational conservation and dominance of uORFs between *Drosophila* species

Given that the translation of uORFs is a crucial determinant of their functional impact, we first characterized the translational profiles of uORFs in *D. melanogaster* and *D. simulans* to explore the evolutionary roles of uORFs in translation regulation. We focused on canonical uORFs that initiate with an ATG start codon in the 5′ UTR and terminate with a stop codon (TAA, TAG, or TGA). Because the ATG start codon is the defining feature of a canonical uORF and tends to be more conserved than its downstream sequence (*Zhang et al., 2021*), we defined uORF conservation based on the presence of the ATG start codon in the 5′ UTR of *D. melanogaster* and its orthologous positions in *D. simulans*, regardless of differences in the stop codon. Using this criterion, we identified 18,412 canonical uORFs with conserved start codons between the two species. Additionally, we identified 2789 canonical uORFs specific to *D. melanogaster* and 2440 canonical uORFs specific to *D. simulans*. The TE values of the conserved uORFs were highly correlated between the two species across all developmental stages and tissues examined, with Spearman correlation coefficients (*rho*) ranging from 0.478 to 0.573 (*Figure 2A*). In contrast, TE of CDSs exhibited a significantly higher correlation between the two species in the corresponding samples compared to that of uORFs, with Spearman's *rho* ranging from 0.588 to 0.806 (p=0.002, Wilcoxon signed-rank test; *Figure 2A*). This observation is consistent with our simulation results, which indicate that uORFs experience greater translational fluctuations than their downstream CDSs. Notably, conserved uORFs exhibited significantly higher TEs compared to species-specific uORFs in both species. The median TE of conserved uORFs was 1.62 times that of non-conserved uORFs in *D. simulans*, while the corresponding ratio in *D. melanogaster* was 1.52 (*Figure 2B*).

In *D. melanogaster*, 7259 (52.2%) genes had no uORFs, 2687 (19.3%) had a single uORF, and 3961 (28.5%) contained multiple uORFs. Among genes with multiple uORFs, we defined the uORF with the highest TE as the dominant uORF for that gene, as TE is one of the most relevant metrics for assessing uORF function (*Johnstone et al., 2016*; *Zhang et al., 2021*). The median TE of the dominant uORF was 4.84 times that of the second-highest uORF within the same gene in *D. melanogaster*, and the corresponding ratio was 5.21 times in *D. simulans* (*Figure 2—figure supplement 1*). To assess the consistency of this dominance across different tissues and developmental stages, we identified 3072 multiple-uORF genes in *D. melanogaster* with at least one translated uORF (TE >0.1 in at least five stages/tissues). Of these, 569 genes consistently used the same dominant uORF across the measured samples, significantly higher than the number expected under randomness (5 genes, 95% confidence interval: 1–10) based on shuffling the TEs of uORFs 1000 times (*Figure 2—figure supplement 2*). This trend was also observed in *D. simulans* and persisted under different thresholds for defining 'translated uORFs' (*Figure 2—figure supplement 2*). These results suggest that genes with multiple uORFs tend to retain the same dominant uORF across developmental stages, indicating that the dominant uORF may serve as the key translational regulator of the downstream CDS. Moreover, we found that the dominant uORFs showed a higher proportion of conserved uATGs than the other translated uORFs (median proportion 82.5% vs 78.8%, p<0.001; *Figure 2C*). Additionally, the absolute values of TE fold-change ($|\log_2 TE\text{-}FC|$) between the two species were, on average, 23.2% smaller for dominant uORFs than for other conserved uORFs (*Figure 2D*), suggesting that dominant uORFs are more likely

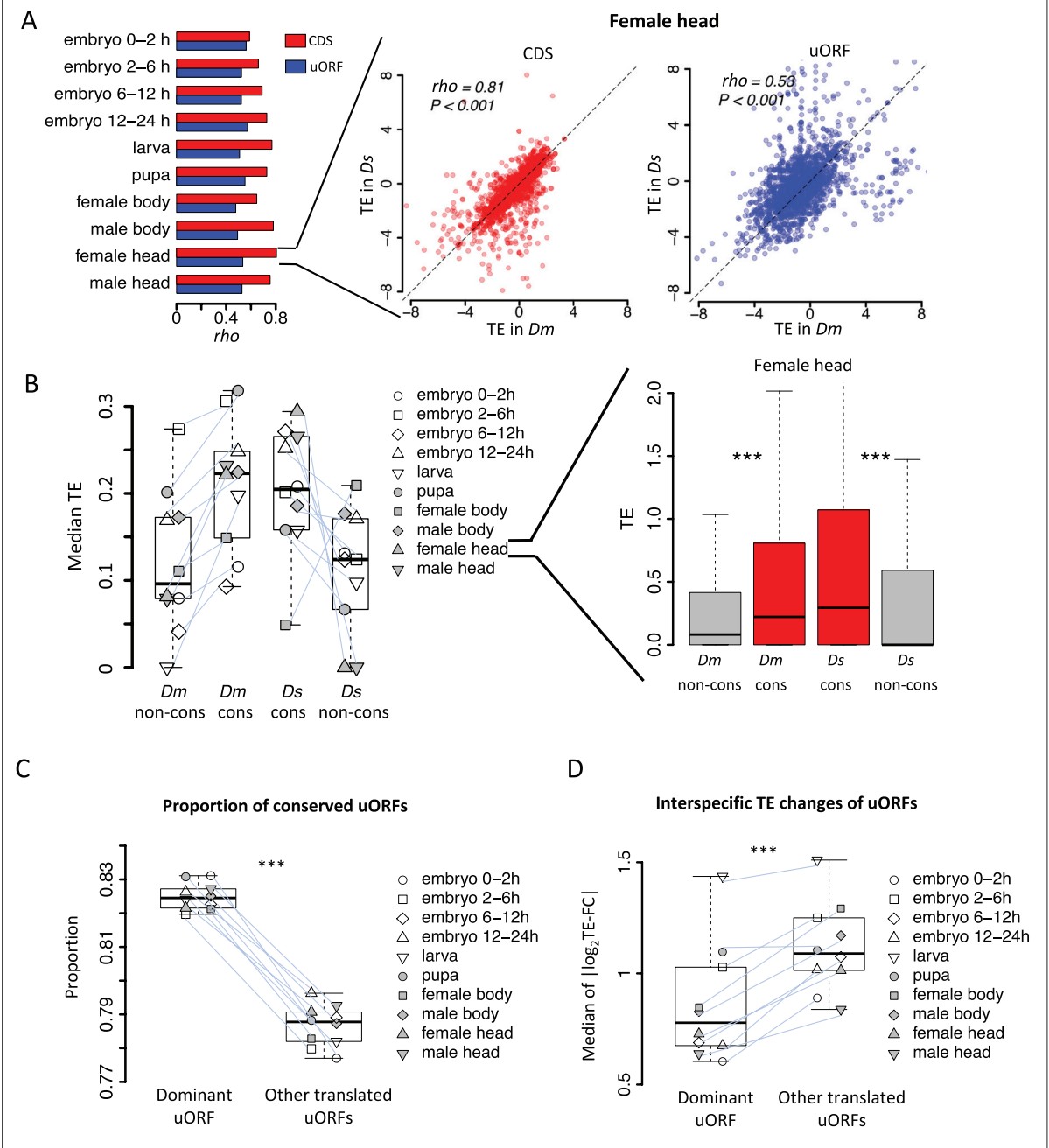

**Figure 2.** Conservation and translation of uORFs between *D. melanogaster* and *D. simulans*. (**A**) Spearman's correlation coefficients (*rho*, represented by the bars) of TEs between *Dm* (*D. melanogaster*) and *Ds* (*D. simulans*) for CDS (red) and uORFs (blue). All *P*-values for Spearman's correlation are less than 0.001. The p value for the comparison between *rho* values of CDSs TE and uORFs TE is 0.002 (Wilcoxon signed-rank test). Data for the female head sample is shown as an example in the right panel. The x- and y-axes represent the TEs in *Dm* and *Ds*. (**B**) The median of TE of conserved and species-specific uORFs in each sample. Each dot represents the median TE of a sample for a specific uORF class. Data from the female head sample is shown as an example in the right panel. p values were obtained from Wilcoxon rank-sum tests. ***, p<0.001. (**C**) Fraction of conserved uORFs among dominant uORFs and other translated uORFs in each sample. The paired samples in *Dm* and *Ds* were linked together. The p value was obtained by the paired Wilcoxon signed-rank test. ***, p<0.001. (**D**) Absolute values of the interspecific TE fold changes (log$_2$TE-FC) of dominant uORFs and the other translated uORFs in each sample. The paired samples in *Dm* and *Ds* were linked together. The median value of each sample is shown. The p value was obtained via the paired Wilcoxon signed-rank test. ***, p<0.001. Data from the female head sample were used as an example in the right panel.

The online version of this article includes the following source data and figure supplement(s) for figure 2:

**Source data 1.** Raw numerical data underlying the figures.

**Figure supplement 1.** The dominant uORF showed the highest TE than other uORFs with the same gene.

*Figure 2 continued on next page*

*Figure 2 continued*

**Figure supplement 1—source data 1.** Raw numerical data underlying the figures.

**Figure supplement 2.** Observed and expected numbers of genes sharing dominant uORFs in all samples.

**Figure supplement 2—source data 1.** Raw numerical data underlying the figures.

under stronger stabilizing selection. These findings suggest that, in genes with multiple uORFs, the dominantly translated uORF may play a more important role in regulating CDS translation than the other uORFs.

## uORFs buffer interspecific translational divergence of CDSs

To investigate the relationship between translational changes in uORFs and their downstream CDSs across species, we analyzed the translation efficiency (TE) of uORFs and corresponding downstream CDSs in *D. melanogaster* and *D. simulans*. Consistent with our simulations that the translation of a uORF is tightly linked to that of downstream CDS, uORFs exhibited a modest, yet statistically significant, positive correlation with the TE of their downstream CDSs across all samples analyzed (p<0.001, Spearman's correlation) (*Figure 3A*). We then compared the interspecific TE change of a uORF ($\beta_u = TE_{uORF,sim}/TE_{uORF,mel}$) with that of its corresponding CDS ($\beta_C = TE_{CDS,sim}/TE_{CDS,mel}$) between *D. melanogaster* and *D. simulans*. We found $\beta_u$ is significantly positively correlated with $\beta_C$ across all samples (p-values <0.001, Spearman's correlation; *Figure 3—figure supplement 1*). These results align well with our simulations, which showed that fluctuations in translational factors (such as ribosomes) influence both uORF and CDS translation in the same direction (*Figure 1—figure supplements 8 and 9*).

While the direction of TE changes for uORFs and CDSs tends to be consistent, our simulations suggest that the magnitude of TE changes in CDSs is generally smaller than that in uORFs, due to the buffering effect of uORF (*Figure 1—figure supplements 10 and 11*). To validate this, we first identified uORFs and CDSs with significant interspecific TE differences by assessing whether $\beta_u$ or $\beta_C$ significantly deviated from 1, using an established statistical framework (*Zhang et al., 2018a*). This analysis uncovered 1151–4189 CDSs with significant interspecific TE changes (FDR <0.05; *Table 1*), with genes involved in development, morphogenesis, and differentiation being significantly enriched during embryonic stages (*Figure 3—figure supplement 2*). Conversely, genes related to metabolism, response to stimuli, and signaling were enriched in larval, pupal, and adult stages (*Figure 3—figure supplement 2*). Additionally, we identified 144–1193 uORFs with significant TE differences between species, accounting for approximately 1–15% of expressed uORFs (*Table 1*). Note that due to their shorter length and generally lower TE, uORFs had considerably lower read counts than CDSs, limiting the statistical power to detect significant interspecific TE differences for uORFs. This trend consistently holds whether analyzing all expressed uORFs (*Figure 3—figure supplement 3A*) or only highly expressed genes (*Figure 3—figure supplement 3B*). Thus, the fewer uORFs showing significant TE divergence likely reflects lower read counts and statistical sensitivity rather than reduced translational variability relative to CDSs. In fact, the absolute values of $\log_2$(fold change) of TE for uORFs between *D. melanogaster* and *D. simulans* were significantly greater than those observed for corresponding CDSs across all samples (p<0.001, Wilcoxon signed-rank test; *Figure 3B*), suggesting that the magnitude of TE changes in CDSs is generally smaller than that in uORFs, due to the buffering effect of uORF.

We further quantitatively compared the magnitude of interspecific TE changes between uORFs and their corresponding CDSs using a previous method (*Zhang et al., 2018a*). We defined $\gamma = \beta_c/\beta_u$ for a uORF-CDS pair within the same mRNA and tested whether $\gamma$ was significantly different from 1 to identify pairs with differential TE changes between uORFs and CDSs (*Figure 3—figure supplement 4*). When $\gamma$ <1 and $\beta_u$ >1, or $\gamma$ >1 and $\beta_u$ <1, it indicates that the TE change for the CDS is smaller than that for the uORF (*Figure 3—figure supplement 4*). Among CDS-uORF pairs where $\beta_u$ >1, nearly all (8–487) showed a significant $\gamma$ < 1 in each sample, except for one pair from the pupal stage where $\gamma$ >1 (*Table 1*). This suggests that the magnitude of TE changes in CDSs was generally smaller than in uORFs when uORF TE increased, and vice versa when uORF TE decreased ($\beta_u$ <1) (*Table 1*). These comparative translatome analyses indicate that uORFs buffered downstream CDS translation changes during the evolutionary divergence of *D. melanogaster* and *D. simulans*.

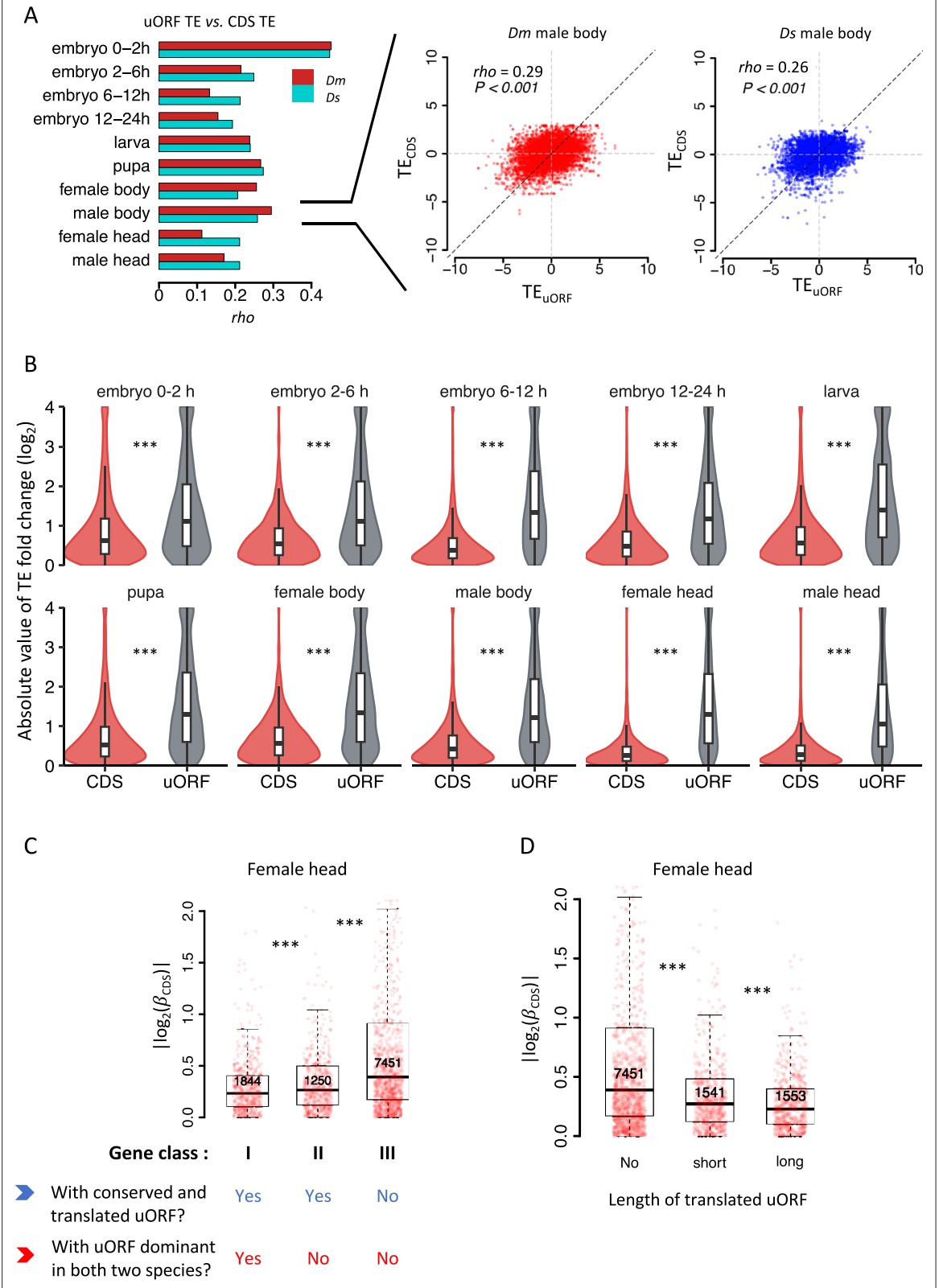

**Figure 3.** uORFs reduce CDS translational divergence between *D. melanogaster* and *D. simulans*. (**A**) The correlation of uORF TEs and the corresponding CDS TEs in 10 samples of *Dm* (*D. melanogaster*) and *Ds* (*D. simulans*). The bars represent Spearman's correlation coefficient (*rho*). In all samples, we obtained both p-values <0.001. Data for the female head sample of *Dm* and *Ds* are shown as examples in the right panel. (**B**) The absolute values of interspecific TE changes for CDS and uORF in each sample between two species. For visualization purposes, all values greater than 4 were

*Figure 3 continued*

assigned a value of 4. ***, p<0.001, Wilcoxon rank-sum test. (**C**) Genes expressed in female heads (mRNA RPKM >0.1 in both species) were classified into three classes according to whether a gene had a conserved and dominantly translated uORF or not. Boxplots showing interspecific CDS TE variability $|log_2(\beta_c)|$ of different gene classes. p values were calculated using Wilcoxon rank-sum tests between the neighboring groups. ***, p<0.001. (**D**) Genes expressed in female heads were classified into three classes according to the length of translated uORFs. Boxplots showing interspecific CDS TE variability $|log_2(\beta_c)|$ of different gene classes. p values were calculated using Wilcoxon rank-sum tests between the neighboring groups. ***, p<0.001.

The online version of this article includes the following source data and figure supplement(s) for figure 3:

**Source data 1.** Raw numerical data and statistical analysis underlying the figures.

**Figure supplement 1.** The positive correlation of interspecific TE changes between uORFs and CDSs.

**Figure supplement 1—source data 1.** Raw numerical data underlying the figures.

**Figure supplement 2.** Gene ontology analysis of the genes with $log_2(\beta_2) \neq 0$ in each stage and tissue.

**Figure supplement 3.** Reads count distribution of uORFs and CDSs.

**Figure supplement 3—source data 1.** Raw numerical data underlying the figures.

**Figure supplement 4.** The scheme illustrating the calculation of $\beta_u$, $\beta_c$ and $\gamma$.

**Figure supplement 5.** Conserved and dominantly translated uORFs showed the stronger buffering effect.

**Figure supplement 5—source data 1.** Raw numerical data underlying the figures.

**Figure supplement 6.** Longer uORFs showed the stronger buffering effect.

## uORF buffering is influenced by its conservation, dominance, and length

To investigate how the conservation level and translation patterns of uORFs influence their buffering capacity on CDS translation, we categorized genes expressed in each pair of samples into

**Table 1.** Numbers of genes showing different magnitudes of TE changes between uORFs and CDS at the interspecific level.

| Sample types | # of expressed uORFs * | $\beta_u \neq 1$ (%) † | # of expressed CDSs * | $\beta_c \neq 1$ (%) † | uORF-CDS pairs with $\beta_u$>1 | | | uORF-CDS pairs with $\beta_u$<1 | | |
|---|---|---|---|---|---|---|---|---|---|---|
| | | | | | Total | $\gamma$>1 | $\gamma$<1 | Total | $\gamma$>1 | $\gamma$<1 |
| 0–2 hr embryo | 7704 | 1193 (15.49) | 7934 | 4189 (52.80) | 770 | 0 | 366 | 423 | 69 | 0 |
| 2–6 hr embryo | 7822 | 567 (7.25) | 9795 | 3063 (31.27) | 249 | 0 | 63 | 318 | 135 | 0 |
| 6–12 hr embryo | 10,400 | 1040 (10.00) | 10,643 | 2,924 (27.47) | 973 | 0 | 641 | 67 | 22 | 1 |
| 12–24 hr embryo | 11,365 | 535 (4.71) | 11,158 | 3537 (31.70) | 454 | 0 | 234 | 81 | 8 | 2 |
| Larva | 10,008 | 464 (4.64) | 11,831 | 3554 (30.04) | 110 | 0 | 26 | 354 | 177 | 0 |
| Pupa | 12,309 | 635 (5.16) | 12,209 | 4087 (33.48) | 136 | 1 | 15 | 499 | 210 | 0 |
| Male body | 10,894 | 197 (1.81) | 12,284 | 2432 (19.80) | 122 | 0 | 29 | 75 | 4 | 0 |
| Male head | 10,904 | 144 (1.32) | 10,447 | 1151 (11.02) | 119 | 0 | 31 | 25 | 5 | 0 |
| Female body | 9809 | 340 (3.47) | 11,002 | 3605 (32.77) | 279 | 0 | 17 | 61 | 3 | 0 |
| Female head | 10,935 | 332 (3.04) | 10,545 | 1270 (12.04) | 324 | 0 | 152 | 8 | 3 | 0 |

*Only uORFs and CDSs with an mRNA RPKM >0.1 in both *D. melanogaster* and *D. simulans* were considered in each sample pair in the analysis.

† $\beta_u = TE_{uORF,sim}/TE_{uORF,mel}$ is the fold change of $TE_{uORF}$ in *D. simulans* relative to *D. melanogaster* for each sample. $\beta_u = TE_{CDS,sim}/TE_{CDS,mel}$ is the fold change of $TE_{CDS}$ in *D. simulans* relative to *D. melanogaster* for each sample. $\gamma = \beta_c/\beta_u$. For each CDS–uORF pair, either $\beta_u > 1$ and $\gamma > 1$, or $\beta_u < 1$ and $\gamma > 1$ means that the magnitude of TE change is lower for a CDS than for a uORF. The statistical significance of $\beta_c$, $\beta_u$, and $\gamma$ were determined according to an FDR < 0.05

The online version of this article includes the following source data for table 1:

**Source data 1.** Raw data and statistical analysis for uORF TE changes.

**Source data 2.** Raw data and statistical analysis for CDS TE changes.

three classes: Class I, genes with conserved uORFs that are dominantly translated (i.e., exhibiting the highest TE among all uORFs within the same gene) in both *Drosophila* species; Class II, genes with conserved uORFs that are translated in both species but not dominantly translated in at least one; and Class III, the remaining expressed genes. We then compared the absolute values of interspecific TE changes of the CDS ($|\beta_c|$) across these three categories. Significant differences in $|\beta_c|$ were observed, with a consistent hierarchy of Class I < II < III across all pairs of samples (*Figure 3C*, *Figure 3—figure supplement 5*). On average, Class I genes exhibited an average of 8.18% and 23.8% lower $|\beta_c|$ values compared to Class II and Class III, respectively. This indicates that conserved and dominantly translated uORFs exert a stronger buffering effect on CDS translation.

To further validate the simulation results suggesting that longer uORFs have a stronger buffering effect (*Figure 1C*, *Figure 1—figure supplements 6 and 7*), we divided genes expressed in each pair of samples into three groups: those without translated uORFs (No), those with short uORFs (short, total length below the median), and those with long uORFs (long, total length above the median). Consistently, longer uORFs were associated with stronger buffering effects on CDS translation across all pairs of samples (*Figure 3D*, *Figure 3—figure supplement 6*). Specifically, genes with longer uORFs showed 12.7% and 26.5% lower $|\beta_c|$ values compared to genes with short uORFs or no uORFs, respectively.

Overall, these findings underscore that the buffering capability of a uORF is positively correlated with its conservation level, translation dominance, and length.

## uORFs buffer translational fluctuations during *Drosophila* development

Gene expression undergoes dynamic changes during *Drosophila* development, with significant alterations in the translation program to meet developmental demands, including shifts in ribosome loading rates (*Kronja et al., 2014*; *Malzer et al., 2013*; *Qin et al., 2007*). Therefore, we extended our analysis to investigate the role of uORFs in buffering these translational fluctuations, hypothesizing that uORFs could mitigate them during development. According to our hypothesis, if a gene has a translated uORF in *D. melanogaster* but not in its orthologous gene in *D. simulans*, then the translation of this gene is likely more stable across developmental stages in *D. melanogaster* than its ortholog in *D. simulans*, and vice versa.

To test this hypothesis, we compared the CV of CDS TE across 10 developmental stages in *D. melanogaster* and in *D. simulans*, respectively. Genes with translated uORFs (TE >0.1 in at least one sample) in *D. melanogaster* exhibited significantly 22.5% smaller CVs in *D. melanogaster* than their orthologs lacking these uORFs in *D. simulans*, indicating more stable translation (*Figure 4A*). Consistently, for genes with translated uORFs in *D. simulans* but not in *D. melanogaster*, the CVs were 13.3% lower in *D. simulans* compared to their orthologs lacking these uORFs in *D. melanogaster* (*Figure 4B*). Consistent results were observed when a uORF is required to be translated (TE >0.1) in all 10 samples (*Figure 4—figure supplement 1A*), in 4 embryonic stages (*Figure 4—figure supplement 1B*), or in 6 stages including embryos, larva, and pupa (*Figure 4—figure supplement 1C*). Moreover, within each species, genes with translated uORFs also showed less variability in CDS TE across developmental stages compared to those without translated uORFs, with a 31.8% reduction in *D. melanogaster* and a 28.9% reduction in *D. simulans* (*Figure 4C*). This effect was consistent across different thresholds for defining 'translated uORFs' (*Figure 4—figure supplement 2*).

These results suggest that uORFs function as translational buffers, reducing gene translation fluctuations during *Drosophila* development.

## Knocking out the uORF of *bcd* increased *bcd* CDS translation in *D. melanogaster*

After verifying the uORF-mediated translational buffering during *Drosophila* evolution and development, we next aimed to directly explore the biological function of these buffering-capable uORFs in vivo. We first applied stringent criteria to identify uORFs with significant buffering effects on CDS translation between *D. melanogaster* and *D. simulans*. Specifically, we looked for (1) uORFs with significant TE changes ($|log_2(\beta_u)| > 1.5$, adjusted p<0.05), (2) negligible changes in its corresponding CDS translation ($|log_2(\beta_C)| < 0.05$, adjusted p>0.05), and (3) a significant difference between the magnitude of these changes ($|log_2(\gamma)| > 1.5$, adjusted p<0.05). We identified 131 uORF-CDS pairs in 103 genes that meet these criteria in at least one stage/tissue (*Supplementary file 3*), with a majority of the genes

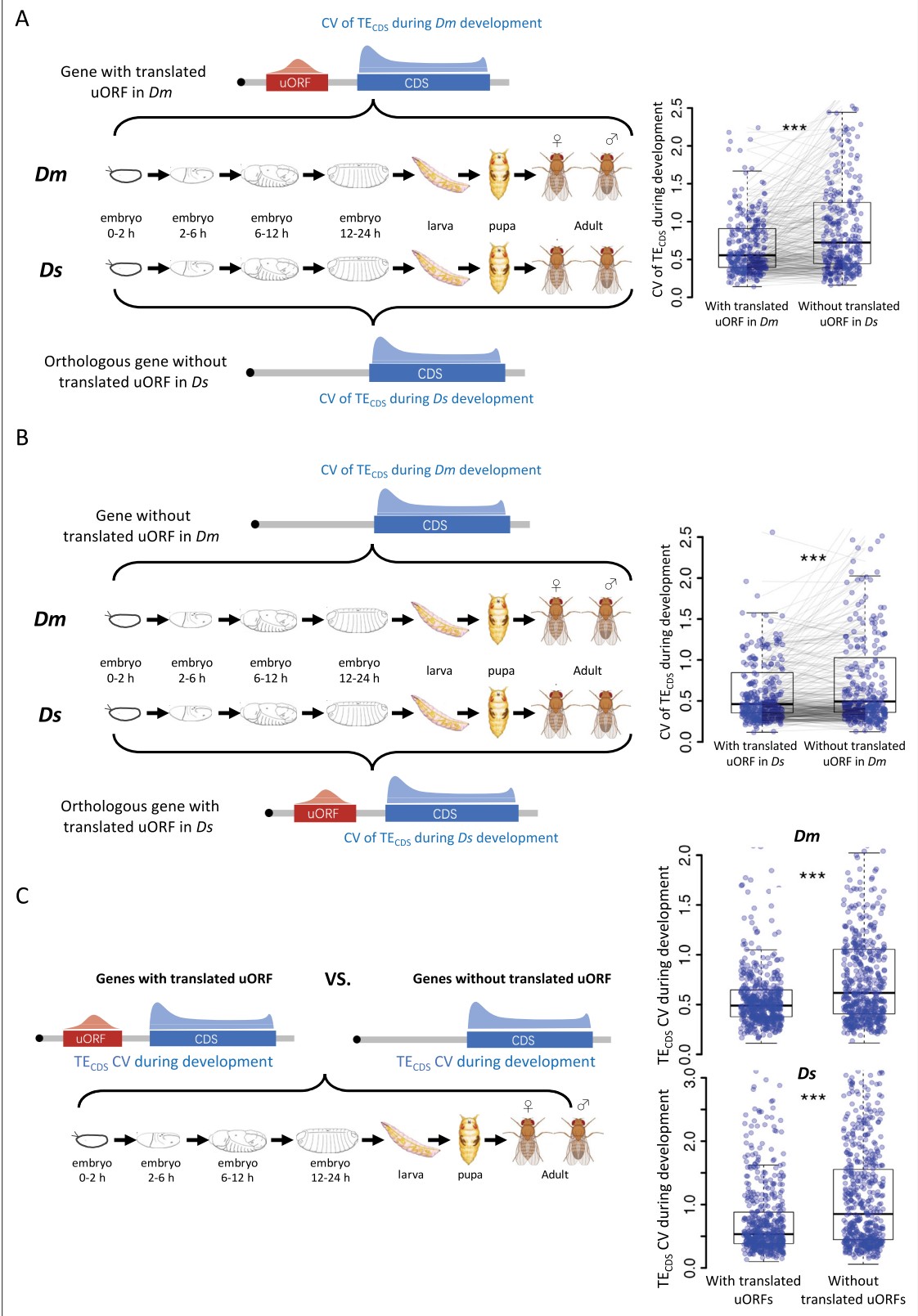

**Figure 4.** uORFs could reduce CDS translational fluctuation during *Drosophila* development. (**A**) The CV of TE$_{CDS}$ across 10 *Dm* (*D. melanogaster*) samples and 10 *Ds* (*D. simulans*) samples. The selected gene with uORFs translated (TE >0.1) in at least one *Dm* sample but its homologous gene without translated uORF in *Ds* samples. Each pair of dots linked by a gray line represents a pair of homologous genes in *Dm* and *Ds*. ***, p<0.001, Wilcoxon signed-rank test. (**B**) The CV of TE$_{CDS}$ across 10 *Dm* samples and 10 *Ds* samples. The selected gene with uORFs translated (TE >0.1) in at

*Figure 4 continued on next page*

*Figure 4 continued*

least one *Ds* sample but its homologous gene without translated uORF in *Dm* samples. Each pair of dots linked by a gray line represents a pair of homologous genes in *Dm* and *Ds*. ***, p<0.001, Wilcoxon signed-rank test. (**C**) Within each *Drosophila* species, the CV of TE$_{CDS}$ of genes with translated uORFs compared to genes without the translated uORFs. The p values are obtained by the Wilcoxon rank-sum test. ***, p<0.001.

The online version of this article includes the following source data and figure supplement(s) for figure 4:

**Source data 1.** Raw numerical data underlying the figures.

**Figure supplement 1.** uORFs reduce CDS translational fluctuation during *Drosophila* development under different cutoffs on defining 'translated uORFs'.

**Figure supplement 1—source data 1.** Raw numerical data underlying the figures.

**Figure supplement 2.** uORFs could reduce CDS translational variation during *Drosophila* development under different cutoffs on defining 'translated uORFs'.

**Figure supplement 2—source data 1.** Raw numerical data underlying the figures.

(67%, 69 out of 103) from embryonic stages (*Figure 5—figure supplement 1A*), suggesting a crucial role for uORFs in maintaining translational stability during early development. Among these genes, one notable case is the *bicoid* (*bcd*) gene, a master regulator of anterior-posterior axis patterning during early embryogenesis (*Berleth et al., 1988*; *Driever and Nüsslein-Volhard, 1988a*; *Driever and Nüsslein-Volhard, 1988b*). The *bicoid* gene contains a 4-codon uORF (excluding the stop codon) in its 5′ UTR, and branch length score (BLS) analysis (*Zhang et al., 2021*) showed that the start codon (uATG) of the *bcd* uORF is highly conserved across the *Drosophila* phylogeny, with a BLS of 0.90 (on a scale from 0 to 1, where higher values indicate greater conservation; *Figure 5A*, *Figure 5—figure supplement 1B*). Ribo-Seq data revealed that in 0–2 hr embryos, the TE of the *bcd* uORF varied more than threefold, while the TE of the CDS was virtually the same between *D. melanogaster* and *D. simulans* (*Figure 5B* and *Supplementary file 3*). This suggests that the *bcd* uORF significantly buffers translation during early development.

To investigate the regulatory role of the *bcd* uORF, we used CRISPR-Cas9 to knock out its start codon in *D. melanogaster*, generating two mutant homozygotes (uKO1/uKO1 and uKO2/uKO2) with a genetic background matched to that of the wild-type (WT; *Figure 6A*, *Figure 6—figure supplement 1*). To determine whether these mutations enhanced *bcd* CDS translation, we performed ribosome fractionation followed by qPCR (*Abdelmohsen et al., 2014*; *Panda et al., 2016*; *Panda et al., 2017*), comparing *bcd* mRNA levels in polysome and monosome fractions (P-to-M ratio) from 0 to 2 hr embryos of uKO1/uKO1, uKO2/uKO2, and WT flies (*Figure 6B*). A larger P-to-M ratio means more mRNAs are enriched in the polysome fractions and bound by more ribosomes, thus indicative of higher translation efficiency. P-to-M ratios were higher in the mutants compared to WT at 29 °C (*Figure 6C*), with a similar but less pronounced trend at 25 °C, where the difference between uKO2/uKO2 and WT was not statistically significant (*Figure 6C*). These findings, along with the known impact of temperature on gene expression and phenotypic plasticity (*Amourda et al., 2018*; *Lu et al., 2021*), suggest that the regulatory function of the uORF and overall translation efficiency are temperature-sensitive, highlighting a complex interplay between environmental conditions and gene regulation. To further confirm the *bcd* uORF's regulatory function, we conducted dual-luciferase reporter assays. The 5′ UTR from *bcd* mutant (uKO1 or uKO2) or WT was cloned into a *Renilla* luciferase reporter construct (*Figure 6D*). Luciferase activity was significantly higher in uKO1 and uKO2 compared to the WT 5′ UTR, confirming the repressive role of the *bcd* uORF.

### *bcd* uORF mutants show wide transcriptomic alteration during *Drosophila* embryogenesis

Since Bcd regulates the expression of many zygotic genes (*Berleth et al., 1988*; *Driever and Nüsslein-Volhard, 1988a*; *Driever and Nüsslein-Volhard, 1988b*), we anticipated that the increased translation of *bcd* resulting from disrupting the uORF would influence *Drosophila* transcriptomes and phenotypes. To verify this notion, we performed RNA sequencing on embryos from WT and uKO2/uKO2 flies at four developmental stages (0–2 hr, 2–6 hr, 6–12 hr, and 12–24 h) under both 25°C and 29°C conditions, using two biological replicates (*Supplementary file 4* and *Figure 6—figure supplement 2*). Differential expression analysis revealed widespread alterations in gene expression between WT and mutant embryos, with the number of differentially expressed genes (DEGs) increasing over

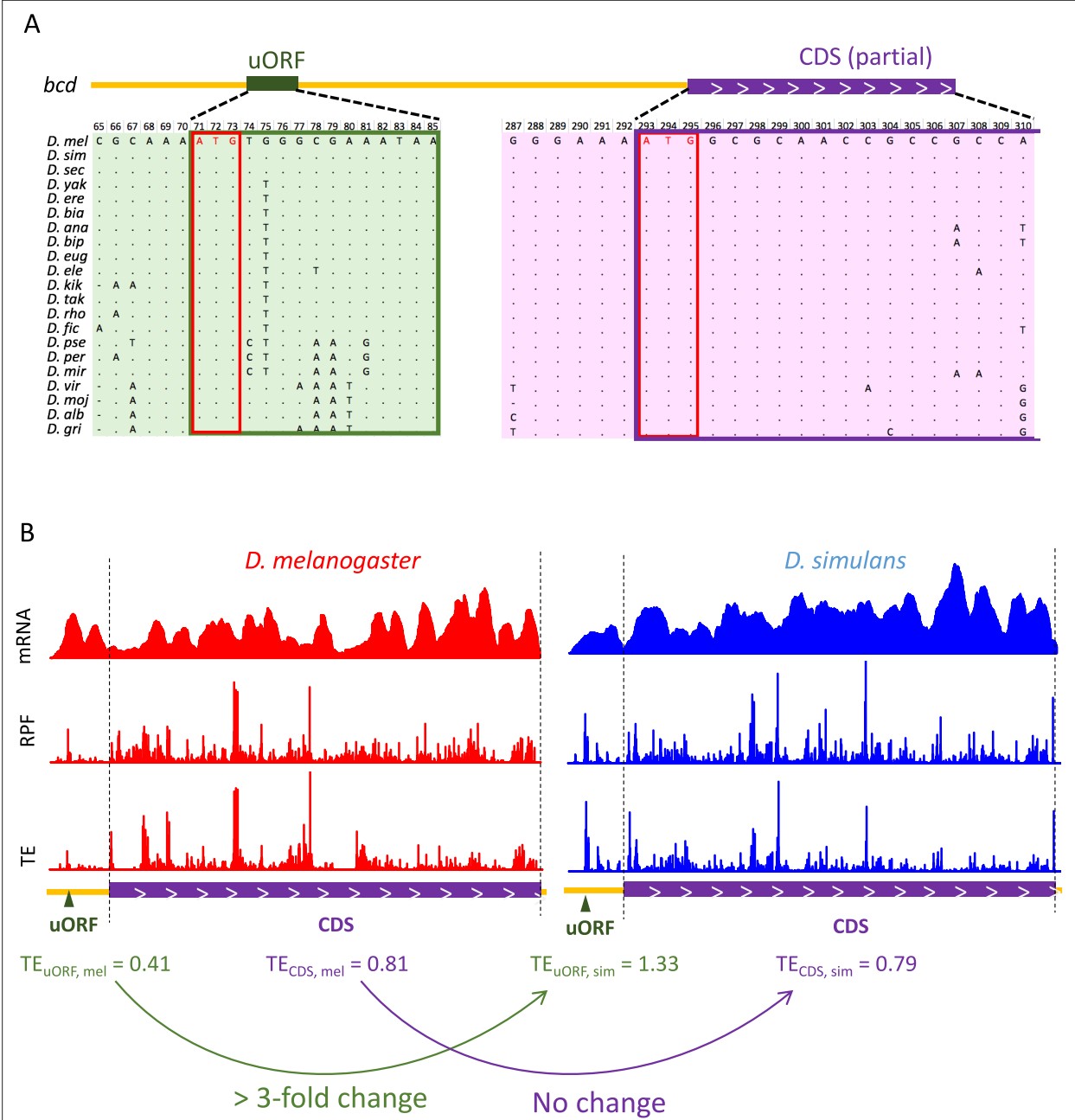

**Figure 5.** The strong buffering effect of the *bcd* uORF on CDS translation between the two *Drosophila* species. (**A**) Multiple sequence alignment of the *bcd* uORF and partial CDS in *D. melanogaster* and 20 other *Drosophila* species. The uORF and CDS are boxed in green and purple, respectively. The start codons of the uORF and CDS are boxed in red. (**B**) The coverage of mRNA-Seq (top), Ribo-Seq (middle), and TEs (bottom) of the *bcd* uORF and CDS in 0–2 hr embryos of *D. melanogaster* (red) and *D. simulans* (blue). The uORF and CDS are denoted at the lower panel with dark green triangles and purple boxes, respectively. The two dashed lines mark the CDS region. The uORF TE, CDS TE and their interspecific changes were labeled at the bottom.

The online version of this article includes the following source data and figure supplement(s) for figure 5:

**Source data 1.** Raw numerical data underlying the figures.

**Figure supplement 1.** Screening of genes containing uORFs exhibiting strong translational buffering effects.

**Figure supplement 1—source data 1.** Raw numerical data underlying the figures.

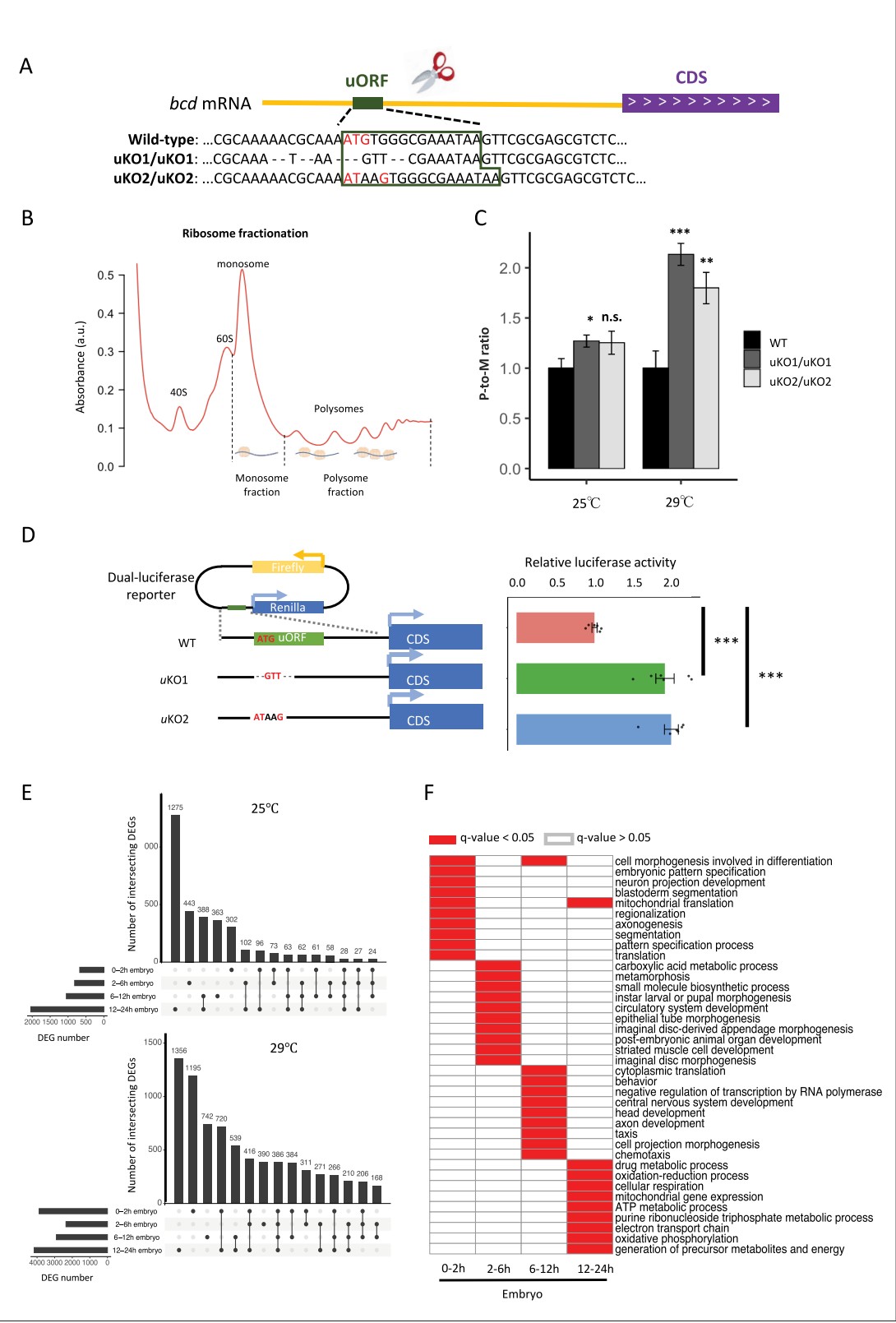

**Figure 6.** Knocking out the *bcd* uORF increases CDS translation and perturbs the transcriptome during *D. melanogaster* embryogenesis. (**A**) Genotypes of WT and two uORF knock-out strains (uKO1 and uKO2) generated by CRISPR-Cas9 technology. The uORF is boxed in dark green, and the red ATG represents the start codon of the uORF in the *D. melanogaster* genome. (**B**) Two ribosome fractions (monosome and polysome) of 0–2 hr embryos were separated in a sucrose density gradient. Relative RNA abundance in the monosome and polysome fractions was quantified by real-time quantitative

*Figure 6 continued on next page*

*Figure 6 continued*

PCR. (**C**) P-to-M ratio of *bcd* mRNA (*bcd* mRNA abundance in polysome fraction/*bcd* mRNA abundance in monosome fraction) at 25 °C (left) and 29 °C (right). The P-to-M ratios of mutants were normalized to WT controls at 25 °C and at 29 °C, respectively. Error bars represent the S.E. of six biological replicates. Asterisks indicate statistical significance (*, p<0.05; **, p<0.01; ***, p<0.001; n.s., p>0.05). (**D**) Dual-luciferase assay for *bcd* WT uORF and mutated uORF. The reporter structures of the WT and uORF mutants are illustrated on the left. The uORF mutant sequence was the same as that in the fly mutant created with CRISPR-Cas9 technology. The relative activity of *Renilla* luciferase was normalized to that of firefly luciferase. Error bars represent the S.E. of six biological replicates. Asterisks indicate statistical significance (***, p <0.001). (**E**) The number of DEGs in each stage and their intersection with each other at 25 °C (top) and 29 °C (bottom). (**F**) Gene ontology analysis of DEGs at 29 °C in each stage. The biological process (BP) terms with q-values <0.05 in each stage are indicated in red, and others are indicated in white.

The online version of this article includes the following source data and figure supplement(s) for figure 6:

**Source data 1.** Raw numerical data underlying the figures.

**Figure supplement 1.** Overview of homozygous uORF-KO mutant screening.

**Figure supplement 2.** Heatmap of sample-to-sample distances of RNA-Seq libraries.

**Figure supplement 2—source data 1.** Raw numerical data underlying the figures.

**Figure supplement 3.** The ratio of Bcd target genes among DEGs in *bcd*-uORF-KO embryos.

**Figure supplement 3—source data 1.** Raw numerical data underlying the figures.

**Figure supplement 4.** The correlation of expression changes (log$_2$FoldChange) between RNA-Seq (y-axis) and RT-qPCR (x-axis) of 20 *bcd* targets in four embryo stages in uKO2/uKO2 mutant compared to WT.

**Figure supplement 4—source data 1.** Raw numerical data underlying the figures.

developmental time and at higher temperature (*Figure 6E*). At 25 °C, we identified 674, 817, 1047, and 2041 DEGs in 0–2 hr, 2–6 hr, 6–12 hr, and 12–24 hr embryos, respectively; while at 29 °C, we detected 3884, 2358, 2901, and 4164 DEGs in the corresponding stages (*Figure 6E*). The majority of DEGs were stage-specific, with only a small fraction consistently differentially expressed across all four stages. Functional enrichment analysis of the DEGs revealed distinct biological pathways affected at each stage, including cell morphogenesis and pattern specification in 0–2 hr embryos, metabolic processes and tissue development in 2–6 hr and 6–12 hr embryos, and mitochondrial respiration in 12–24 hr embryos (*Figure 6F*). Notably, direct targets of Bcd (*Li et al., 2008*) were significantly enriched among the DEGs in three out of the four stages, with the exception of 2–6 hr embryos (*Figure 6—figure supplement 3*). RT-qPCR validation of 20 target genes of Bcd confirmed the reliability of the RNA-seq differential expression analysis (*Figure 6—figure supplement 4*). Together, these findings demonstrate that disruption of the *bcd* uORF leads to widespread transcriptional changes during *Drosophila* development, affecting processes ranging from embryogenesis to postembryonic metabolism.

## *bcd* uORF mutants display decreased hatching rates and starvation resistance

Given the widespread transcriptome alterations, we anticipated phenotypic abnormalities in the *bcd* uORF mutants. As expected, both uKO1/uKO1 and uKO2/uKO2 mutants exhibited significantly lower hatching rates compared to WT [p=1.4 × 10⁻⁶ and 7.9 × 10⁻⁶, respectively, Wilcoxon rank-sum test (WRST); *Figure 7A*]. At 25 °C, uKO1/uKO1 mutants produced fewer offspring than WT flies (p<0.001, WRST, *Figure 7B*). Given that *bcd* is a maternal gene, we expected reciprocal crosses between uKO1/uKO1 mutants and WT flies to produce different outcomes. Indeed, crossing uKO1/uKO1 males with WT females resulted in offspring numbers similar to those from WT crosses, while crossing uKO1/uKO1 females with WT males yielded offspring numbers comparable to crosses between uKO1/uKO1 mutants (*Figure 7B*). This verified that the reduction in offspring is due to maternal defects in the uKO1/uKO1 mutants. Similar patterns were observed for uKO2/uKO2 mutants at 25 °C (*Figure 7B*). Notably, the fecundity reduction in uORF-KO mutants was more pronounced at 29 °C (*Figure 7C*). Furthermore, crossing uKO1/uKO1 and uKO2/uKO2 mutants also produced significantly fewer progeny than WT flies (*Figure 7C*), ruling out genetic background or off-target effects. Collectively, these data demonstrate that disrupting the *bcd* uORF significantly impaired hatchability and fertility.

We also found that both uKO1/uKO1 and uKO2/uKO2 female mutants perished significantly faster than WT flies under starvation conditions (*Figure 7D*). Males showed similar tendencies, although the difference was not statistically significant for uKO1/uKO1 mutants (*Figure 7D*). These data suggest that the knockout of *bcd* uORF diminished starvation resistance in adults, likely due to embryogenesis

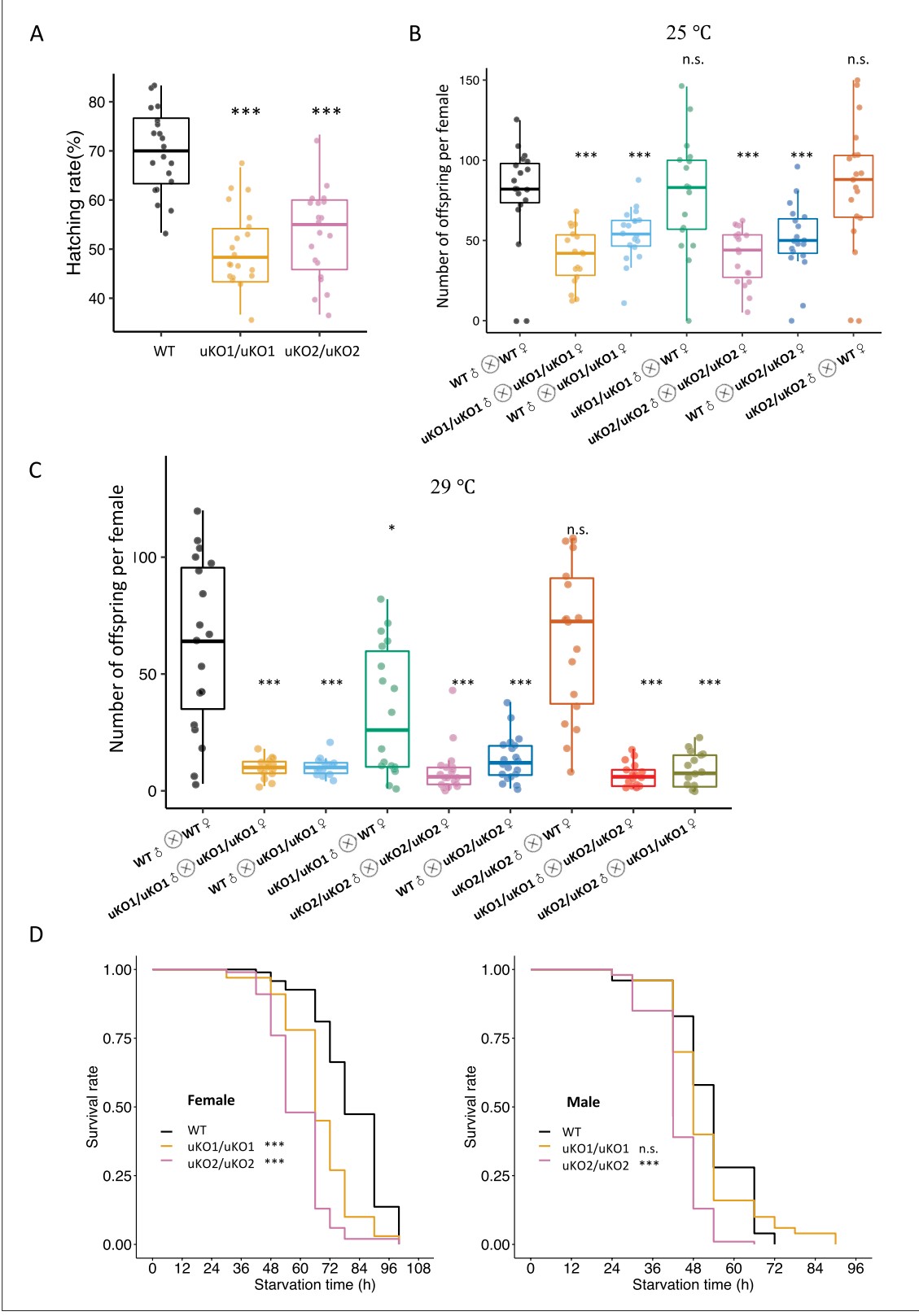

**Figure 7.** Knockout of the *bcd* uORF reduces offspring number and starvation resistance. (**A**) Comparison of the hatching rates (%) of mutant and WT offspring (n=20, Wilcoxon rank-sum test; ***, p<0.001). (**B**) The offspring number per maternal parent in different crosses over 10 days at 25 °C. Asterisks indicate significant differences between various crosses and crosses of WT females with WT males (n=20, Wilcoxon rank-sum test; *, p<0.05; **, p<0.01; ***, p<0.001; n.s., p>0.05). The different crosses were denoted as the x-axis labels. (**C**) The offspring number per maternal parent in different crosses

*Figure 7 continued on next page*

*Figure 7 continued*

over 10 days at 29 °C. (**D**) Survival curves of WT and mutant adult flies of females (left) and males (right) under starvation conditions. The black line represents the WT, the red line represents the uKO1/uKO1 mutant, and the blue line represents the uKO2/uKO2 mutant. Asterisks indicate significant differences compared to the WT. (n=200, log-rank test; ***, p<0.001; n.s., p>0.05).

The online version of this article includes the following source data for figure 7:

**Source data 1.** Raw numerical data underlying the figures.

abnormalities induced by the *bcd* uORF deletion, even in those that successfully developed to adulthood.

## Conservation of uORF-mediated translational buffering in primates

To explore the generality of uORF-mediated translational buffering across evolutionary clades, we analyzed previously published transcriptome and translatome data from three tissues (brain, liver, and testis) in humans and macaques (**Wang et al., 2020**). We identified 33,680 canonical uORFs in humans and 29,516 in macaques, with 24,385 conserved between the two species. Despite the larger number of uORFs in primates compared to *Drosophila* due to differences in genome size and gene number, the median TE of conserved uORFs was 1.79 times that of non-conserved uORFs in humans, and the corresponding ratio was 3.43 in macaques (**Figure 8A**, **Figure 8—figure supplement 1**). TEs of uORFs were positively correlated between humans and macaques across all tissues (p<0.001, **Figure 8B**). Additionally, significant positive correlations were observed between the TEs of uORFs and their corresponding coding sequences (CDSs) in all tissues (**Figure 8—figure supplement 2**). Although interspecific TE divergence of uORFs ($\beta_u$) and CDSs ($\beta_C$) were positively correlated (p<0.001, **Figure 8C**), uORFs generally exhibited larger divergence (**Supplementary file 5**). Notably, longer uORFs showed stronger buffering effects on CDS translation, reducing interspecific TE divergence by 12.9% and 38.0% compared to genes with short or no uORFs (**Figure 8D**, **Figure 8—figure supplement 3**).

As in *Drosophila*, we categorized expressed human genes into three classes based on uORF conservation and translation: Class I, genes with conserved and dominantly translated uORFs in both humans and macaques; Class II, genes with conserved uORFs translated in both species but not dominantly in at least one; and Class III, the remaining expressed genes (**Figure 8E**, **Figure 8—figure supplement 4**). Consistent with findings in *Drosophila*, significant differences in $|\beta_c|$ between humans and macaques were observed in the order of Class I < II < III, with Class I genes showing 9.8% and 17.1% lower $|\beta_c|$ values compared to Class II and Class III genes, respectively (**Figure 8E**, **Figure 8—figure supplement 4**). These similarities between primates and *Drosophila*—two clades that diverged over 700 million years ago (**Hedges et al., 2006**) —suggest that uORF-mediated translational buffering is a widespread mechanism for stabilizing gene translation across evolutionary clades.

We also analyzed matched mRNA-Seq and Ribo-Seq data from 69 human lymphoblastoid cell lines (**Battle et al., 2015**; **Lappalainen et al., 2013**) to test whether uORFs buffer against translational variability across different individuals within humans. Genes with translated uORFs exhibited, on average, 8.65% lower CVs in CDS TE across individuals than genes without translated uORFs (**Figure 8F**), with longer uORFs showing stronger buffering effects (**Figure 8—figure supplement 5**). Collectively, these findings suggest that uORFs play a crucial role in reducing translational variability both across evolutionary clades and within species.

## Discussion

Translational control is vital for maintaining protein homeostasis and cellular activities (**Sherman and Qian, 2013**; **Stein and Frydman, 2019**). However, the mechanisms underlying the conservation of protein abundance across species remain largely unknown (**Signor and Nuzhdin, 2018**; **Vogel, 2013**). In this study, we extended the ICIER model and conducted simulations to explore uORFs' regulatory roles in buffering translation during evolution. Our simulations demonstrated that uORF translation reduces variability in downstream CDS translation, with buffering capacity positively correlated with uORF translation efficiency, length, and number. Comparative translatome analyses across developmental stages of two *Drosophila* species provided evidence that uORFs mitigate interspecific differences in CDS translation. Similar patterns were observed between humans and macaques, and

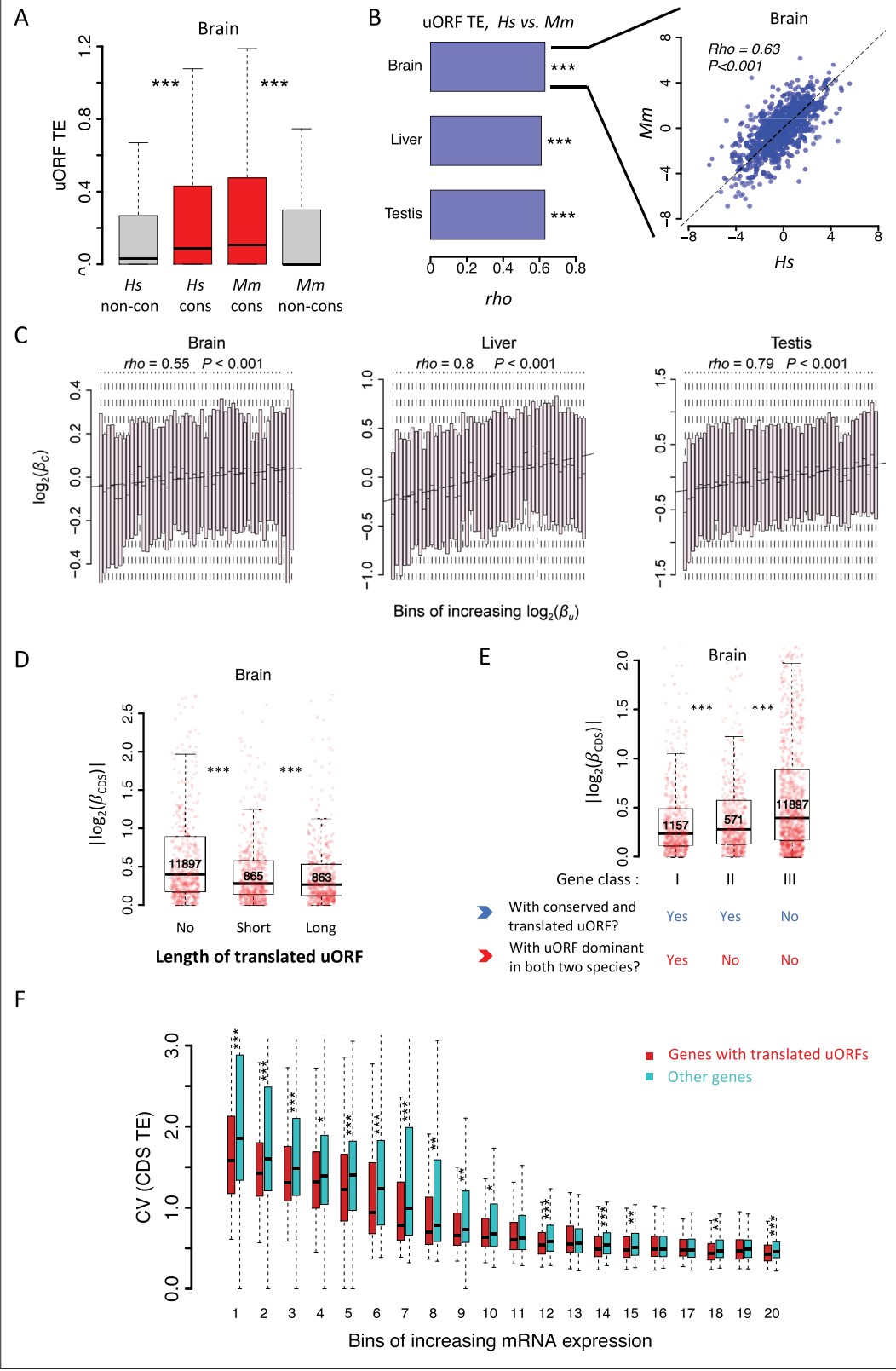

**Figure 8.** uORFs function as translational buffers in primates. (**A**) Boxplots showing the TEs of conserved and species-specific uORFs between *Hs* (*H. sapiens*) and *Mm* (*M. mulatta*). Data for the brain is shown as an example. Wilcoxon rank-sum tests. ***, p<0.001. (**B**) Spearman's correlation coefficient (*rho*) of uORFs' TE between humans and macaques. The *rho* values in the brain, liver, and testis were shown as bar plots. ***, p<0.001. Data for the

*Figure 8 continued on next page*

*Figure 8 continued*

brain is shown as an example in the right panel. (**C**) Correlation between interspecific uORF TE changes ($\log_2 \beta_u$) and corresponding CDS TE changes ($\log_2 \beta_C$) in three tissues. The x-axis was divided into 50 equal bins with increasing $\beta_u$. (**D**) Genes expressed in brains were classified into three classes according to the total length of translated uORFs. Boxplots showing interspecific CDS TE variability $|log_2(\beta_c)|$ of different gene classes. p values were calculated using Wilcoxon rank-sum tests between the neighboring groups. \*\*\*, p<0.001. (**E**) Genes expressed in brains (mRNA RPKM >0.1 in both species) were classified into three classes according to whether a gene had a conserved and dominantly translated uORF (TE >0.1) in both species or not. Boxplots showing interspecific CDS TE variability $|log_2(\beta_c)|$ of different gene classes. p values were calculated using Wilcoxon-rank sum tests between the neighboring groups. \*\*\*, p<0.001. (**F**) Boxplot showing the coefficients of variation (CVs) of CDS TE among the 69 lymphoblastoid cell lines (LCLs). Expressed genes (mean mRNA RPKM >0.1) were divided into 20 bins with increased mRNA expression levels. In each bin, the genes were divided into two fractions according to whether the gene had a translated uORF or not. Wilcoxon rank-sum tests. \*, p<0.05; \*\*, p<0.01; \*\*\*, p<0.001.

The online version of this article includes the following source data and figure supplement(s) for figure 8:

**Source data 1.** Raw numerical data underlying the figures.

**Figure supplement 1.** Boxplots showing the TEs of conserved and species-specific uORFs in the liver and testis of *H.*

**Figure supplement 1—source data 1.** Raw numerical data underlying the figures.

**Figure supplement 2.** Correlations of uORF TE and corresponding CDS TE in 3 tissues of *H.*

**Figure supplement 2—source data 1.** Raw numerical data underlying the figures.

**Figure supplement 3.** Boxplots showing interspecific CDS TE variability $|log_2(\beta_{CDS})|$ for three gene groups according to the length of their translated uORFs (No, short, long) in the liver (left) and testis (testis).

**Figure supplement 3—source data 1.** Raw numerical data underlying the figures.

**Figure supplement 4.** Boxplots showing interspecific CDS TE variability $|log_2(\beta_{CDS})|$ for different gene classes in the liver and testis.

**Figure supplement 4—source data 1.** Raw numerical data underlying the figures.

**Figure supplement 5.** Left panel: Boxplot showing the coefficients of variation (CVs) of CDS TEs among 69 human lymphoblastoid cell lines (LCLs) for three gene groups.

**Figure supplement 5—source data 1.** Raw numerical data underlying the figures.

---

between human individuals, suggesting that uORF-mediated buffering is an evolutionarily conserved mechanism in animals. Additionally, in vivo experiments showed that knocking out the *bcd* uORF led to aberrant embryogenesis and altered starvation resistance, with more pronounced phenotypic effects at higher temperatures, supporting the role of uORFs in fine-tuning translation and phenotypic outcomes.

While the prevailing consensus on uORFs' functions is that they repress the translation of downstream CDS by sequestering ribosomes (*Andreev et al., 2018*; *Calvo et al., 2009*; *Hinnebusch et al., 2016*; *Johnstone et al., 2016*; *Kozak, 1989*; *Morris and Geballe, 2000*; *Young and Wek, 2016*), recent studies have suggested uORFs might play essential roles in stabilizing protein expression (*Andreev et al., 2018*; *Bottorff et al., 2022*; *Wu et al., 2022*). While these studies have primarily focused on the role of uORFs in single-gene cases or under stress conditions, our work expands this knowledge by demonstrating that uORFs act as a general mechanism for buffering translational variability on a genome-wide scale across multiple species and developmental stages. Our findings also suggest that uORF-mediated translational buffering is not merely a response to environmental stress but an evolutionarily conserved mechanism integral to maintaining protein homeostasis across species. Organisms can buffer phenotypic variation against environmental or genotypic perturbations through 'canalization' (*Waddington, 1942*). The discovery that uORFs are crucial for *Drosophila* development links them to canalization, providing evidence that uORFs reduce interspecific CDS translation differences and enhance phenotypic stability. While previous studies primarily focused on the stabilization of transcriptional levels (*Denby et al., 2012*; *Ebert and Sharp, 2012*; *Jarosz et al., 2010*; *Lu et al., 2023*; *Payne and Wagner, 2015*) or protein levels (*Alon, 2007*; *Jarosz et al., 2010*; *Somogyvári et al., 2022*; *Wu et al., 2022*; *Zabinsky et al., 2019*), our study opens new avenues for exploring how organisms maintain phenotypic robustness at the translational level through molecular

mechanisms like uORF-mediated buffering. This study lays the groundwork for future research into the evolutionary and functional roles of uORFs, particularly in the context of canalization and adaptive evolution.

Previous studies have shown that a significant fraction of fixed uORFs in the populations of *D. melanogaster* and humans were driven by positive Darwinian selection (*Zhang et al., 2018a*; *Zhang et al., 2021*), suggesting active maintenance through adaptive evolution rather than purely neutral or deleterious processes. While uORFs have traditionally been recognized for their capacity to attenuate translation of downstream CDSs, accumulating evidence now underscores their critical role in stabilizing gene expression under fluctuating cellular and environmental conditions (*Andreev et al., 2018*; *Bottorff et al., 2022*; *Wu et al., 2022*). Whether the favored evolutionary selection of uORFs acts primarily through their role in translational repression or translational buffering remains a compelling yet unresolved question, as these two functions are inherently linked. Indeed, highly conserved uORFs tend to be translated at higher levels, resulting not only in stronger inhibition of CDS translation (*Calvo et al., 2009*; *Johnstone et al., 2016*; *Zhang et al., 2021*), but also in a more pronounced buffering effect, as demonstrated in this study. This buffering capacity of uORFs potentially provides selective advantages by reducing fluctuations in protein synthesis, thus minimizing gene-expression noise and enhancing cellular homeostasis. This suggests that selection may favor uORFs that contribute to translational robustness, a hypothesis supported by findings in yeast and mammals showing that uORFs are significantly enriched in stress-response genes and control the translation of certain master regulators of stress responses (*Andreev et al., 2015*; *Lawless et al., 2009*; *Mueller and Hinnebusch, 1986*; *Vattem and Wek, 2004*). Our study suggests that translational buffering, rather than translational repression alone, can also drive evolutionary selection favoring uORFs, although it remains challenging to empirically disentangle these functions. Future comparative genomic analyses, coupled with experimental approaches such as ribosome profiling and functional mutagenesis, will be crucial in elucidating the precise evolutionary forces driving uORF conservation and adaptation.

The original ICIER model was devised to elucidate how a single uORF can confer resistance to global translation inhibition on its host gene during stress conditions (*Andreev et al., 2018*). It is proposed that an elongating ribosome (80S) causes downstream 40S subunits to dissociate from the mRNA after collision. Our revised model expands this scope to include bidirectional dissociation, where an 80S ribosome can release both upstream and downstream 40S subunits, reflecting recent findings that 80S primarily causes upstream 40S dissociation during 40S/80S collisions (*Bottorff et al., 2022*). Mechanistically, increasing the loading rate of 40S ribosomes onto the mRNA ($R_{in}$) elevates the density of 40S and 80S ribosomes scanning the uORF, thereby raising the likelihood of 40S-80S collisions and subsequent 40S dissociation. This leads to a reduced flow of 40S ribosomes reaching the CDS relative to the initial increase in 40S at the 5′ end of the mRNA and uORF. Conversely, a decrease in the 40S ribosome loading rate reduces the density of scanning ribosomes, diminishing the probability of 40S-80S collisions and 40S dissociation, and partially restoring the flow of 40S ribosomes to the CDS. Overall, the 40S-80S collision mechanism and subsequent 40S dissociation at the uORF moderate downstream CDS translation less than changes in ribosome loading at the 5′ end of the mRNA or uORF. The revised model underscores the analogy of uORFs as 'molecular dams', adeptly modulating the stream of ribosomes and cushioning downstream CDS translation against fluctuations (*Figure 9A*). Additionally, our study broadens its perspective to account for the potential interactive effects of multiple uORFs within a gene, recognizing the prevalent clustering of these regulatory elements and their combinatorial influence on translational control. In essence, our study presents a nuanced and comprehensive expansion of the ICIER model, paving the way for a deeper understanding of uORFs as pivotal elements in the maintenance of protein synthesis stability under variable translational conditions. However, we realize the existence of alternative mechanisms governing uORF function, such as those found in traditional repression models (*Wu et al., 2022*) or ribosome queuing and reinitiation strategies (*Bottorff et al., 2022*). As such, further computational studies are needed to dissect and understand the full spectrum of uORF-mediated translational regulation.

Ribosome slowdown or stalling on mRNA due to rare codons (*Bottorff et al., 2022*; *Ivanov et al., 2018*; *Lin et al., 2019*; *Meijer and Thomas, 2003*) or nascent blocking peptides (*Ito and Chiba, 2013*; *Lovett and Rogers, 1996*; *Vilela and McCarthy, 2003*; *Wei et al., 2012*) frequently triggers ribosome collisions genome-wide (*Han et al., 2020*; *Meydan and Guydosh, 2020*; *Zhao et al., 2021*). Such collisions, especially among elongating 80S ribosomes, often activate ribosome quality

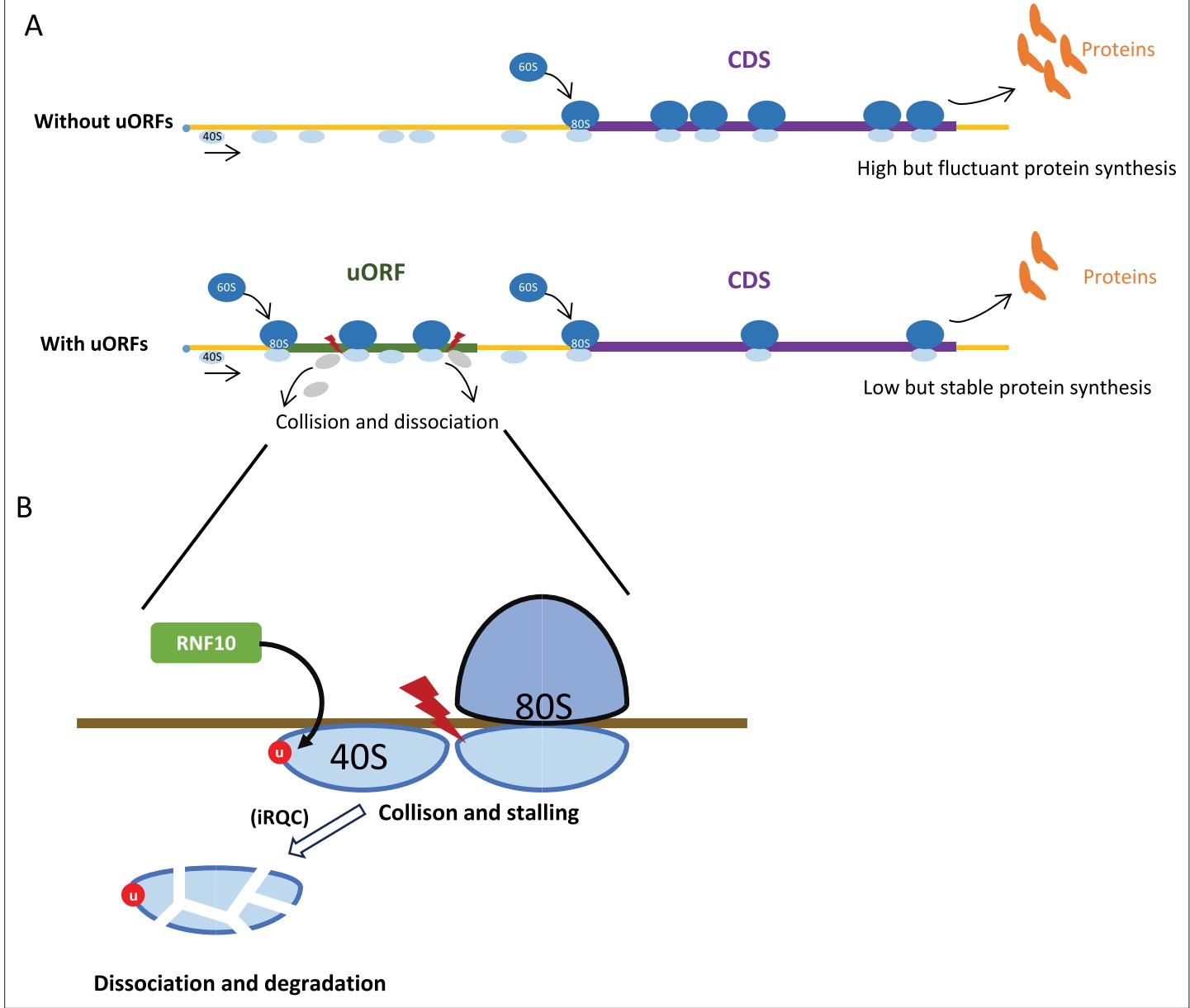

**Figure 9.** Model illustrating the uORF-mediated translational buffering. (**A**) uORFs can reduce and stabilize the downstream CDS translation through 'ribosomes collision and dissociation' mechanism in an mRNA containing uORFs. In contrast, the CDS translation is higher and more fluctuant in an mRNA without uORFs. (**B**) The collided and stalled 40S subunit dissociates from mRNA, is ubiquitinated by the E3 ubiquitin ligase RNF10, and degraded by the initiation ribosome quality control (iRQC) pathway.

control (RQC) pathways that recognize collision interfaces on the 40S subunit, leading to ribosomal subunit dissociation and degradation (*Goldman et al., 2021*; *Juszkiewicz et al., 2020*; *Simms et al., 2019*). In mammals, ZNF598 specifically identifies collided ribosomes to initiate ubiquitin-dependent protein and mRNA quality control pathways (*Geng et al., 2024*; *Juszkiewicz et al., 2018*; *Narita et al., 2022*; *Sundaramoorthy et al., 2017*; *Ugajin et al., 2023*). Analogously, yeast employs Hel2-mediated ubiquitination of uS10, initiating dissociation via the RQC-trigger complex (RQT; *Best et al., 2023*). Furthermore, the human RQT (hRQT) complex recognizes ubiquitinated ribosomes and induces subunit dissociation similarly to yeast RQT (*Hashimoto et al., 2020*). However, transient ribosome collisions can evade RQC by promoting resumed elongation through mechanical force provided by trailing ribosomes, thereby mitigating stalling (*Madern et al., 2025*). Beyond 80S collisions, evidence increasingly highlights a distinct collision type involving scanning 40S subunits or pre-initiation (43S)

complexes. Recently, an initiation RQC pathway (iRQC) targeting the small ribosomal subunit (40S) has been described, particularly involving collisions between scanning 43S complexes or between stalled 43S and elongating 80S ribosomes (*Figure 9B*; *Ford et al., 2025*; *Garshott et al., 2021*). During iRQC, E3 ubiquitin ligase RNF10 ubiquitinates uS3 and uS5 proteins, resulting in 40S degradation (*Ford et al., 2025*). This mechanism aligns closely with our ICIER model, proposing collision-driven 43S dissociation in the 5′ UTRs. Future studies exploring these mechanisms in greater detail will clarify how uORFs modulate translational regulation through buffering effects.

The *bcd* gene, crucial for *Drosophila* embryogenesis, has been well studied (*Cho et al., 2005*; *Driever and Nüsslein-Volhard, 1988a*; *Driever and Nüsslein-Volhard, 1988b*; *Dubnau and Struhl, 1996*; *Hannon et al., 2017*; *Singh et al., 2022*; *Struhl et al., 1989*), but its uORF remains underexplored. Our study showed that deleting the *bcd* uORF led to abnormal phenotypes and reduced fitness, underscoring the importance of uORF-mediated translation. These effects were more severe under heat stress (29°C) than at normal temperatures (25°C), suggesting uORFs play a vital role in buffering phenotypic variation. We demonstrated that the *bcd* uORF represses CDS translation using sucrose gradient fractionation followed by qPCR—an approach that directly measures translation efficiency while minimizing confounding from RNA/protein degradation. However, detecting Bcd protein levels with antibodies across developmental stages or conditions in the mutants and wild-type controls would provide an even stronger validation of our model and should be explored in future studies. Rescue experiments, typically used to confirm that phenotypic changes are due to specific genetic edits, faced challenges in our study and previous uORF-KO experiments in animals (*Miyake et al., 2023*; *Wethmar et al., 2010*) and plants (*Si et al., 2020*; *Xing et al., 2020*; *Xue et al., 2023*; *Zhang et al., 2018b*). We altered the start codon of the *bcd* uORF to minimize effects on the 5′ UTR, without impacting the coding sequence, making traditional rescue experiments impractical. To control for genetic background variations, we backcrossed two *bcd* uORF-KO mutant strains with *w1118* flies for nine generations, using *w1118* as a control. Both uORF-KO mutants (uKO1/uKO1 and uKO2/uKO2) consistently showed reduced progeny, ruling out background or off-target effects. Crosses between the two mutants produced fewer offspring than WT crosses, confirming that the reduced progeny was due to the uORF deletions, not genetic background. As *bcd* is maternally inherited, we expected defective phenotypes only in offspring from mutant females crossed with wild-type males. This was confirmed: uKO1/uKO1 males crossed with wild-type females had normal offspring, while uKO1/uKO1 females crossed with wild-type males had reduced offspring, similar to mutant crosses. Similar results were observed for uKO2/uKO2 mutants. These findings confirm that disrupting *bcd* uORFs significantly reduces hatchability and fertility, effects not attributable to genetic background. Our results highlight the critical role of *bcd* uORFs in development and fecundity and provide a basis for future studies on uORF regulation in other genes across species.

Taken together, this study demonstrates the role of uORF-mediated translational buffering in mitigating variability in gene translation during species divergence and development, using large-scale comparative transcriptome and translatome data. Our work addresses gaps in understanding the stabilization of gene expression at the translational level and offers new insights into the evolutionary and functional significance of uORFs.

## Materials and methods

**Key resources table**

| Reagent type (species) or resource | Designation | Source or reference | Identifiers | Additional information |
|---|---|---|---|---|
| Gene (*D. melanogaster*) | *bicoid* | FlyBase | FBgn0000166 | N/A |
| Strain, strain background (*D. simulans*) | Sim4 (*D. simulans*) | This study | N/A | Used to generate all *D. simulans* libraries |
| Genetic reagent (*D. melanogaster*) | y,v; attp40{nos-Cas9} | Tsinghua Fly Center | TH00788 | Host line injected with sgRNA plasmid |
| Genetic reagent (*D. melanogaster*) | *w1118* | Bloomington *Drosophila* Stock Center | BDSC: 3605 | Used for outcrossing and wild-type control |

*Continued on next page*

*Continued*

| Reagent type (species) or resource | Designation | Source or reference | Identifiers | Additional information |
|---|---|---|---|---|
| Genetic reagent (*D. melanogaster*) | *uKO1/uKO1* | This study | N/A | Bcd-uORF-knockout mutant |
| Genetic reagent (*D. melanogaster*) | *uKO2/uKO2* | This study | N/A | Bcd-uORF-knockout mutant |
| Cell line (*D. melanogaster*) | S2 cells | Thermo Fisher Scientific | R69007 | Mycoplasma-free, regularly tested using PCR-based assay |
| Biological sample (*D. simulans*) | Embryos, larva, pupa, bodies, heads | This study | N/A | Used for mRNA-seq and Ribo-seq |
| Biological sample (*D. melanogaster*) | Embryos, larva, pupa, bodies, heads | *Zhang et al., 2018a* | N/A | Used for mRNA-seq and Ribo-seq |
| Recombinant DNA reagent | pU6B | Tsinghua Fly Center | N/A | sgRNA-expression vector |
| Recombinant DNA reagent | psiCHECK-2 | Promega | C8021 | Used for dual-luciferase reporter assay |
| Commercial assay or kit | PrimeScript II 1st Strand cDNA Synthesis Kit | Takara | Cat# 6210 A | Used for cDNA synthesis |
| Commercial assay or kit | PowerUp SYBR Green Master Mix | Thermo Fisher Scientific | N/A | Used for RT-qPCR |
| Commercial assay or kit | Dual-Luciferase Reporter Assay System | Promega | E1980 | For measuring Renilla and firefly luciferase activity |
| Chemical compound, drug | SuperaseIn RNase inhibitor | Ambion | Cat# AM2694 | RNase inhibitor |
| Chemical compound, drug | Proteinase inhibitor cocktail | Roche | N/A | Proteinase inhibitor |
| Chemical compound, drug | Lipofectamine 3000 | Thermo Fisher Scientific | L3000001 | transfection of plasmids into S2 cells |
| Software, algorithm | R | R Foundation for Statistical Computing | https://www.r-project.org | Version 4.2.3; packages used include ggplot2, DESeq2, etc. |
| Software, algorithm | Python | Python Software Foundation | https://www.python.org | Version 3.10 |
| Software, algorithm | Cutadapt 3.0 | https://cutadapt.readthedocs.io | N/A | Used for adaptor trimming |
| Software, algorithm | Bowtie2 v2.2.3 | https://bowtie-bio.sourceforge.net/bowtie2/index.shtml | N/A | Used for read mapping to contaminant/reference genomes |
| Software, algorithm | STAR | https://github.com/alexdobin/STAR | RRID:SCR_004463 | Used for aligning mRNA-Seq and Ribo-Seq reads to reference genomes |
| Software, algorithm | psite (Plastid) | https://plastid.readthedocs.io | N/A | Used to assign P-sites of ribosome-protected fragments (RPFs) |
| Sequence-based reagent | Bcd-uORF-sgRNA-F | This paper | sgRNA for *Bcd* uORF knock-out | ATCGCAAAAACGCAAAATGT |
| Sequence-based reagent | Bcd-uORF-sgRNA-R | This paper | sgRNA for *Bcd* uORF knock-out | ACATTTTGCGTTTTTGCGAT |
| Sequence-based reagent | Bcd-qPCR-F | This paper | RT−qPCR primers used for *Bcd* | GATGTATCTGGGTGGCTGCT |
| Sequence-based reagent | Bcd-qPCR-F | This paper | RT−qPCR primers used for *Bcd* | CCGAAATGTGGGACGATAAC |
| Sequence-based reagent | Bcd-genotyping-F | This paper | Primers used for Bcd mutant genotyping | GCTTTGCCGTACTGTTCGAT |

*Continued on next page*

*Continued*

| Reagent type (species) or resource | Designation | Source or reference | Identifiers | Additional information |
|---|---|---|---|---|
| Sequence-based reagent | Bcd-genotyping-R | This paper | Primers used for Bcd mutant genotyping | AACTGAAGCTGCGGATGTTG |
| Sequence-based reagent | Rp49-qPCR-F | This paper | RT−qPCR primers used for *rp49* | CACTTCATCCGCCACCAGTC |
| Sequence-based reagent | Rp49-qPCR-R | This paper | RT−qPCR primers used for *rp49* | CGCTTGTTCGATCCGTAACC |

## Modeling the uORF-mediated buffering effect on CDS translation

We adapted the stochastic ICIER framework developed by *Andreev et al., 2018* based on the TASEP model, with several major modifications. First, we concatenated a 500-codon CDS downstream of the uORF. A leaky scanning ribosome from the uORF would initiate translation at the CDS start codon with a probability of $I_{CDS}$. The probability of elongating ribosomes moving along the CDS ($\nu_{EC}$=0.5) in each action was higher than that along uORFs ($\nu_{Eu}$=0.3) in the model settings, considering that uORFs usually encode blocking peptides (*Ito and Chiba, 2013*; *Lovett and Rogers, 1996*; *Luo and Sachs, 1996*; *Vilela and McCarthy, 2003*; *Wei et al., 2012*) or contain stalling codons (*Bottorff et al., 2022*; *Ivanov et al., 2018*; *Meijer and Thomas, 2003*). Second, we recorded the number of elongating ribosomes that completed translation at the stop codon of a CDS ($N_{EC}$) or uORF ($N_{Eu}$) during a given time period and regarded them as proxies for quantifying the CDS or uORF TE. Third, we considered three models for simulating the consequences of scanning ribosomes colliding with elongating ribosomes mentioned above: a downstream dissociation model, an upstream dissociation model, and a double dissociation model.

In detail, the mRNA molecule structure in the simulations consisted of five parts: a 5' leader before the uORF (150 nucleotides), the uORF (ranging from 2 to 100 codons), a segment between the uORF and CDS (150 nucleotides), the CDS (500 codons) and the 3' UTR (150 nucleotides). The $R_{in}$ determined the loading rate of the 40S scanning ribosomes on the mRNA 5'-terminus, and it roughly represented the availability of *trans* translational resources in the cell. We generated two sets of $R_{in}$ values (*Figure 1—figure supplement 1A*) to simulate the variation in the availability of translational resources (40S scanning ribosomes, etc.): (i) 1000 values of $R_{in}$ were generated from a random generator of a uniform distribution, $U$ (0, 0.1); (ii) 1000 values were first generated from a random generator of an exponential distribution, $E$ (1), and then each of the 1000 values was divided by 70 to obtain 1000 values of $R_{in}$. $I_{uORF}$ and $I_{CDS}$ represent the probability of translation at the start codon of a uORF or CDS, respectively. We used different combinations of $I_{uORF}$ and $I_{CDS}$ to test how translational initiation strength influenced the buffering effect of uORFs. $\nu_s$ determined the scanning rate of the 40S ribosome, and $\nu_{Eu}$ and $\nu_{EC}$ determined the elongation rate of the 80S ribosome at a uORF or CDS, respectively. All the parameters used in our simulation are listed in *Supplementary file 1*.

The mRNA molecule was modeled as an array, and the value of each position in the array represented the occupation status of ribosomes along the mRNA molecule. The process of translation was simulated by a series of discrete actions referred to *Andreev et al., 2018*. Upon each action, there was a probability that certain events would occur, including the addition of a 40S ribosome to the 5'-terminus, the transformation of a 40S ribosome into an 80S ribosome at a CDS or uORF start codon, the movement of a 40S ribosome or 80S ribosome to the next codon in the CDS or uORF, etc. More specific operations in each action were described in Appendix 1.

After 1,000,000 actions, we recorded the number of 80S ribosomes that completed translation at the uORF ($N_{Eu}$) and CDS ($N_{EC}$) for each $R_{in}$ input, which was used to represent the protein production rates (i.e. translation rate) of the uORF and CDS, respectively. To test the effects of different factors, we used various combinations of the parameters in *Supplementary file 1* in the simulations. For a given distribution of $R_{in}$ input (1000 values) and the dissociation model, we obtained the corresponding 1000 $N_{EC}$ values and their CV values by calculating the ratio of the standard deviation to the mean of $N_{EC}$.

In the double-uORF simulation, we only adopted a uniform distribution of $R_{in}$ input and the downstream dissociation model. Most parameters and simulation processes were similar to those in the

single-uORF simulation, except that an additional uORF was introduced upstream of the CDS with an initiation probability of $I_{uORF2}$.

## Annotation of uORFs in *D. melanogaster* and *D. simulans*

We downloaded the whole-genome sequence alignment (maf) of *D. melanogaster* (dm6) and 26 other insect species from the University of California Santa Cruz (UCSC) genome browser ([genome.ucsc.edu](genome.ucsc.edu)) (*Rosenbloom et al., 2015*). To improve the genome correspondence between *D. melanogaster* and *D. simulans*, we adopted a newly published reference-quality genome of *D. simulans* (NCBI, ASM438218v1; *Chakraborty et al., 2021*) to replace the original *D. simulans* genome in UCSC maf. We softly masked repetitive sequences in the new genome by RepeatMasker 4.1.1 ([http://www.repeatmasker.org](http://www.repeatmasker.org)) and aligned it to dm6 with lastz (*Chiaromonte et al., 2002*) in runLastzChain.sh following UCSC guidelines. The lastz alignment parameters and the scoring matrix were the same as the parameters of dm6 and droSim1 in UCSC. Chained alignments were processed into nets by the chainNet and netSyntenic programs. The alignment was integrated into the multiple alignments of 27 species with multiz (*Blanchette et al., 2004*).

Based on the genome annotation of *D. melanogaster* (FlyBase r6.04, [https://flybase.org/](https://flybase.org/)), we used the Galaxy platform to parse the multiple sequence alignments of 5′ UTRs in *Drosophila*. The 5′ UTR sequences of each annotated transcript from *D. melanogaster* and the corresponding sequences in *D. simulans* were extracted from the maf. The start codons of putative uORFs were identified by scanning all the ATG triplets (uATGs) within the 5′ UTRs of *D. melanogaster* and *D. simulans*. uATGs that overlapped with any annotated CDS region were removed. The presence or absence of each uATG of *D. melanogaster* was determined at orthologous sites in *D. simulans* based on multiple genome alignments, and vice versa. The conserved uORF was defined as a uORF where its uATG was present in both *D. melanogaster* and the corresponding orthologous positions of *D. simulans*. For each protein-coding gene, we only considered the canonical transcript. For transcripts containing multiple uORFs, we defined the uORF that showed the highest TE as the dominant uORF. The branch length score (BLS) of the uATGs was calculated as we previously described (*Zhang et al., 2021*).

## Fly materials and general raising conditions

The *sim4* strain of *D. simulans* was used to generate all the libraries of *D. simulans* in this study. All flies were raised on standard corn medium and grown in 12 hr light: 12 hr dark cycles at 25 °C for general conditions or at 29 °C for specific experimental design. The samples of embryos at different stages, larva, pupa, bodies, and heads were collected following a previous protocol (*Zhang et al., 2018a*).

## Processing *Drosophila* mRNA-Seq and Ribo-Seq data

Ribo-Seq and matched mRNA-Seq libraries for different developmental stages and tissues of *D. simulans* were constructed as we previously described (*Zhang et al., 2018a*) and sequenced on an Illumina HiSeq-2500 sequencer (run type: single-end; read length: 50 nucleotides) according to the manufacturer's protocol. The 3′ adaptor sequences (TGGAATTCTCGGGTGCCAAGG) were trimmed using Cutadapt 3.0 with default parameters (*Martin, 2011*), and the NGS reads were mapped to the genomes of yeast, *Wolbachia*, *Drosophila* viruses and the sequences of tRNAs, ribosomal RNAs (rRNAs), small nuclear RNAs (snRNAs) or small nucleolar RNAs (snoRNAs) of *D. melanogaster* (FlyBase r6.04) and *D. simulans* (FlyBase r1.3) using Bowtie2 version 2.2.3 (*Langmead and Salzberg, 2012*) with the parameters -p8 −−local -k1. The mapped reads of these genomes/sequences were further removed in the downstream analysis.

After filtering, the mRNA-Seq and Ribo-Seq reads were mapped to the reference genomes of *D. melanogaster* (FlyBase, r6.04) and *D. simulans* (NCBI, ASM438218v1), respectively, using the Spliced Transcripts Alignment to a Reference (STAR) algorithm (*Dobin et al., 2013*). For Ribo-Seq reads, we assigned a mapped RPF (27–34 nucleotides in length) to its P-site using the psite script from Plastid (*Dunn and Weissman, 2016*). The uniquely mapped reads were extracted and then mapped to the CDS of *D. melanogaster* and *D. simulans,* respectively, using STAR (*Dobin et al., 2013*). The P-sites of RPF or mRNA-Seq reads that overlapped with a CDS were counted separately. The reads that were not mapped to CDSs were then mapped to the 5′ UTRs of *D. melanogaster* and *D. simulans*. The P-sites of RPF or mRNA-Seq reads that overlapped with uORFs were counted separately.

The TE of a given uORF or CDS was calculated as the $RPKM_{P-site}/RPKM_{mRNA}$ ratio. For a few uORFs with $RPKM_{mRNA} = 0$ but $RPKM_{P-site} > 0$, a pseudocount of 0.1 was added to both $RPKM_{mRNA}$ and $RPKM_{P-site}$ to avoid dividing a positive value by zero. In each sample, expressed genes were defined as the genes with a CDS $RPKM_{mRNA} > 0.1$ in both species, and translated uORFs were defined as uORFs with a TE > 0.1.

## Testing the statistical significance of the difference in the interspecific TE change between a uORF and its downstream CDS

For this analysis, we adopted the methods developed by *Zhang et al., 2018a*. Briefly, for the samples of female bodies or male bodies of *D. melanogaster* with biological replicates, we obtained the $log_2(TE)$ and SE of the $log_2(TE)$ of a uORF or CDS by contrasting the RPF counts against mRNA-Seq read counts using DESeq2. Then, we fitted the SE values against the normalized mRNA counts and $log_2(TE)$ values using the gam function in the R package mgcv, with a log link to obtain the SE ~mRNA counts + $log_2TE$ functions. For other samples without biological replicates, we estimated the SE of the $log_2(TE)$ for a feature (CDS or uORF) by applying the fitted functions obtained based on the biological replicates of female and male bodies to the observed mRNA counts and $log_2(TE)$ values. We identified uORFs whose TEs differed significantly between the paired samples of *D. melanogaster* and *D. simulans* by testing whether the value obtained for $log_2(\beta_u) = log_2(TE_{uORF,sim}) - log_2(TE_{uORF,mel})$ was significantly different from 0. Based on the SE of the $log_2(TE)$ derived as described above, the SE of the $log_2(\beta_u)$ can be derived as follows:

$$SE\_log_2(\beta_u) = \sqrt{(SE\_log_2TE_{uORF,sim})^2 + (SE\_log_2TE_{uORF,met})^2}$$

Then, we defined $\gamma = \frac{\beta_c}{\beta_u}$ (where $\beta_c = TE_{CDS,sim}/TE_{CDS,mel}$) and tested whether $log_2(\gamma)$ was significantly different from 0 to determine whether the magnitude of interspecific TE changes in CDSs and uORFs was significantly different. We obtained $log_2TE_{uORF,sim}$, $log_2TE_{uORF,mel}$, $log_2TE_{CDS,sim}$, and $log_2TE_{CDS,mel}$ as described above and estimated $SE\_log_2TE_{uORF,sim}$, $SE\_log_2TE_{uORF,mel}$, $SE\_log_2TE_{CDS,sim}$, and $SE\_log_2TE_{CDS,mel}$ based on the biological replicates of female and male bodies of *D. melanogaster*. Finally, $log_2(\gamma)$ also follows a normal distribution with SE denoted as $SE\_log_2(\gamma) =$

$$\sqrt{(SE\_log_2TE_{uORF,sim})^2 + (SE\_log_2TE_{uORF,mel})^2 + (SE\_log_2TE_{CDS,sim})^2 + (SE\_log_2TE_{CDS,mel})^2}$$

As the Wald statistic, $\frac{log_2(\gamma)}{SE\_log2(\gamma)}$, follows a standard normal distribution under the null hypothesis that $log_2(\beta_u)=0$, we calculated the p value as follows: $2 \cdot (1 - \phi(\left|\frac{log_2(\gamma)}{SE\_log2(\gamma)}\right|))$.

uORFs are shorter than CDSs and have substantially lower read counts, reducing statistical power when comparing TE changes. To mitigate this bias, we focused on highly expressed genes, defined as those with both mRNA and RPF RPKM values above the 50th percentile in *D. melanogaster* and *D. simulans*. uORFs and CDSs of these genes were used in the calculation of significant TE change.

## Knocking out a uORF with CRISPR-Cas9 technology in *D. melanogaster*

We searched for possible sgRNA target sites near the uATG start codon of a uORF using the Benchling website (https://www.benchling.com/crispr/) to design optimal single guide RNA (sgRNA) sequences with high specificity and low off-target effects. We then synthesized single-stranded complementary DNAs (ssDNAs) and annealed them to obtain double-stranded DNA (dsDNA), which served as the template for sgRNA expression. The template sequences of the sgRNA used for *bcd* uATG-KO are listed in the Key Resources Table. The dsDNA was then ligated into the BbsI-digested pU6B vector. The pU6B-sgRNA plasmid was purified and injected into the embryos of transgenic Cas9 flies collected within one hour of laying at the Tsinghua Fly Center as described in *Ni et al., 2011*. The injected embryos were kept at 25 °C and 60% humidity until adulthood (G0). The G0 adult flies that hatched from injected embryos were individually crossed with other flies (*y sc v*) to increase the number of offspring. Then, the F1 progeny were crossed with flies carrying an appropriate balancer (*Dr, e/TM3, Sb*). After F2 spawning, the F1 individuals were screened for mutations of interest by genotyping. The primers used for genotyping are listed in the Key Resources Table. The F2 progeny whose parents showed positive genotyping results were then screened for the *yellow* gene to separate the

chromosome carrying *nos-Cas9*. The screened F2 males were crossed with the flies containing the same balancer as above. After F2 genotyping, the progeny (F3) of positive F2 individuals showing the same mutation status were crossed individually to generate homozygous mutants in the F4 generation. The original homozygous mutants were sequentially outcrossed with *w^1118* flies for nine generations to purify the genetic background (**Figure 6—figure supplement 1B**).

## Ribosome fraction analysis by sucrose gradient fractionation and RT–qPCR

The 0–2 hr embryos obtained from *w^1118*, uKO1/uKO1, and uKO2/uKO2 mutants raised at 25°C and 29°C were collected and homogenized in a Dounce homogenizer with lysis buffer [50 mM Tris pH 7.5, 150 mM NaCl, 5 mM MgCl$_2$, 1% Triton X-100, 2 mM dithiothreitol (DTT), 20 U/ml SuperaseIn (Ambion), 0.5 tablets of proteinase inhibitor (Roche), 100 μg/ml emetine (Sigma Aldrich), and 50 μM guanosine 5′-[β,γ-imido]triphosphate trisodium salt hydrate (GMP-PNP; Sigma Aldrich)] at 4 °C. The lysates were clarified by centrifugation at 4 °C and 20,000×*g* for 8 min, and the supernatants were transferred to new 1.5 ml tubes. 10–45% sucrose gradients were prepared in buffer (250 mM NaCl, 50 mM Tris pH 7.5, 15 mM MgCl$_2$, 0.5 mM DTT, 12 U/ml RNaseOUT, 0.5 tablets of protease inhibitor, and 20 μg/ml emetine) using a Gradient Master (Biocomp Instruments) in ULTRA-CLEAR Thinwall Tubes (Beckman Coulter). A sample volume of up to 500 μl was applied to the top of each gradient. After ultracentrifugation with a Hitachi P40ST rotor at 35,000×rpm for 3 hr at 4 °C, the monosome and polysome fractions were collected, flash-frozen in liquid nitrogen, and stored at –80 °C until further use.

The RNA in the monosome and polysome fractions was extracted separately using TRIzol reagent (Life Technologies, Inc) and chloroform (Beijing Chemical Works) following the manufacturer's instructions and was reverse transcribed into cDNA using the PrimeScript II 1st Strand cDNA Synthesis Kit (Takara). RT–qPCR analysis of *bcd* cDNA and its targets was performed using PowerUp SYBR Green Master Mix (Thermo Fisher) following the manufacturer's instructions with *rp49* as an internal control. The primer sequences employed for RT-qPCR are listed in the Key Resources Table. For each sample, the ratio of *bcd* mRNA abundance in the polysome fraction to that in the monosome fraction was calculated as the P-to-M ratio. Six biological replicates were performed for each sample.

## Cell lines

*Drosophila* S2 cells were purchased from Thermo Fisher Scientific (product number: R69007) and their identity was not subsequently confirmed. The cells were regularly tested and found negative for mycoplasma contamination using PCR-based assays.

## Dual-luciferase reporter assays

The wild-type (WT) 5′ UTR of *bcd* was cloned from cDNA by PCR, and uATG mutations were introduced into 5′ UTR using specific amplification primers. The WT and mutated 5′ UTR sequences were ligated into a linearized reporter plasmid (psiCHECK-2 vector, Promega). The whole sequence of all the plasmids was validated by Sanger sequencing.

*Drosophila* S2 cells were cultured in Schneider's Insect Medium (Sigma) plus 10% (by volume) heat-inactivated fetal bovine serum, 100 U/ml penicillin and 100 μg/ml streptomycin (Thermo Fisher) at 25°C without CO$_2$ for 24 hr to reach 1–2×10$^6$ cells/ml before further treatments. Plasmid transfection was conducted with Lipofectamine 3000 (L3000001, Thermo Fisher) according to the supplier's protocol. The *Renilla* luciferase activity associated with WT or uORF-mutated 5′ UTRs was measured according to the manual of the Dual-Luciferase Reporter Assay System (Promega) 32 hr after transfection and was normalized to the activity of firefly luciferase.

## mRNA-Seq in the embryos of mutant and WT flies

We collected 0–2 hr, 2–6 hr, 6–12 hr, and 12–24 hr embryos of *w^1118* and uKO2/uKO2 mutants raised at 25°C and 29°C and conducted mRNA-Seq. Library construction and sequencing with PE150 were conducted by Annoroad on the Illumina Nova6000 platform. Two biological replicates were sequenced for each sample. The clean data were mapped to the reference genome of *D. melanogaster* (FlyBase, r6.04) using STAR (**Dobin et al., 2013**). Reads mapped to the exons of each gene were tabulated with htseq-count (**Anders et al., 2015**). The differentially expressed genes were identified using DESeq2

(*McCarthy et al., 2012*). The Gene Ontology (GO) analyses were conducted using the "clusterProfiler" package in R (*Yu et al., 2012*).

## Measurement of embryo hatchability

We collected embryos from WT and mutant flies and manually seeded them in vials containing standard corn medium at a density of 30 embryos per vial and 20 vials per strain. The embryos were cultivated under standard conditions (60% humidity, 12 hr light: 12 hr dark cycles at 25 °C). We counted the number of pupae in each vial after the completion of pupation and calculated the corresponding hatching rate = $number_{pupa}$/ 30.

## Quantification of offspring number per female fly

Newly hatched virgins were picked out and allowed to mature for two days in separate vials. They were then mated by placing one virgin female with three male flies for two days. After that, each female parent was transferred to a new vial to count the offspring number produced in each 10 day period at 25°C and 29°C. All assays were performed with 20 females per genotype.

## Measurement of starvation resistance in adult flies

We selected 3- to 5-day-old adult males and females and placed them in the starvation medium (1.5% agar), with 10 flies per vial and 10 vials for both males and females from each strain. We made observations every 6 hr or 12 hr to count the number of deaths under starvation conditions until all flies had starved to death. The survival curves were plotted by the ggsurvplot package in R.

## mRNA-Seq and Ribo-Seq data analysis in primates

The mRNA-Seq and Ribo-Seq data from the brains, livers, and testes of humans and macaques were downloaded from reference (*Wang et al., 2020*) with accession number E-MTAB-7247 (ArrayExpress). The mRNA-Seq and Ribo-Seq data of human lymphoblastoid cell lines (LCL) from Yoruba individuals were downloaded from references (*Battle et al., 2015*; *Lappalainen et al., 2013*) under accession numbers GSE61742 and E-GEUV-1. The matched mRNA-Seq and Ribo-Seq libraries of 69 individuals were used.

The uORF annotation and downstream analysis procedures for the human and macaque data were similar to those applied in *Drosophila* as described above. The differential analysis of translational efficiency in humans and macaques was conducted by Xtail (*Xiao et al., 2016*). In each pair of human-macaque samples, expressed genes were defined as the genes with a CDS $RPKM_{mRNA}$ >0.1 in both species. The translated uORFs in a sample were defined as uORFs with a TE >0.1. For the human cell line data, expressed genes were defined as genes with a mean CDS $RPKM_{mRNA}$ >0.1 across the cell lines, and translated uORFs were defined as uORFs with a mean TE >0.1.

## Acknowledgements

We thank Drs. Wei Xie, Weiwei Zhai, and Xionglei He for constructive comments and suggestions. We thank the National Center for Protein Sciences at Peking University for their technical assistance. Some of the analyses were performed on the High-Performance Computing Platform of the Center for Life Sciences. This work was supported by grants from the Ministry of Science and Technology of the People's Republic of China (2022YFE0132000), the Yunnan Provincial Science and Technology Project at Southwest United Graduate School (202302 A0370006), the National Natural Science Foundation of China (32070597) and the Natural Science Foundation of Beijing (5212006). The funders had no role in study design, data collection and interpretation, or the decision to submit the work for publication.

## Additional information

### Funding

| Funder | Grant reference number | Author |
|---|---|---|
| Ministry of Science and Technology of the People's Republic of China | 2022YFE0132000 | Jian Lu |
| Yunnan Provincial Science and Technology Project at Southwest United Graduate School | 202302A0370006 | Jian Lu |
| National Natural Science Foundation of China | 32070597 | Jian Lu |
| Natural Science Foundation of Beijing | 5212006 | Jian Lu |

The funders had no role in study design, data collection and interpretation, or the decision to submit the work for publication.

### Author contributions

Yuanqiang Sun, Resources, Data curation, Formal analysis, Validation, Investigation, Visualization, Methodology, Writing – original draft, Writing – review and editing; Yuange Duan, Data curation, Formal analysis, Validation, Investigation, Visualization, Methodology, Writing – original draft, Writing – review and editing; Peixiang Gao, Formal analysis, Investigation; Chenlu Liu, Data curation, Formal analysis, Investigation; Kaichun Jin, Methodology; Shengqian Dou, Wenxiong Tang, Hong Zhang, Data curation; Jian Lu, Conceptualization, Supervision, Funding acquisition, Validation, Investigation, Writing – original draft, Project administration, Writing – review and editing

### Author ORCIDs

Yuanqiang Sun ⓘ https://orcid.org/0009-0007-6967-2234
Yuange Duan ⓘ https://orcid.org/0000-0003-2311-9859
Chenlu Liu ⓘ https://orcid.org/0000-0002-8993-0145
Jian Lu ⓘ https://orcid.org/0000-0002-4409-1667

Reviewer #1 (Public review): https://doi.org/10.7554/eLife.104074.3.sa1
Reviewer #2 (Public review): https://doi.org/10.7554/eLife.104074.3.sa2
Author response https://doi.org/10.7554/eLife.104074.3.sa3

## Additional files

### Supplementary files

Supplementary file 1. Parameters used in our simulation.

Supplementary file 2. Mapping statistics of Ribo-Seq and matched mRNA-Seq libraries.

Supplementary file 3. Genes with uORFs showing strong evidence of translational buffering.

Supplementary file 4. Mapping statistics of mRNA-Seq libraries for *bcd* uKO2/uKO2 mutant and WT flies.

Supplementary file 5. Numbers of genes showing different magnitudes of TE changes between uORFs and CDS at the interspecific level, *H. sapiens* and *M. mulatta*.

MDAR checklist

### Data availability

All deep-sequencing data generated in this study, including single-ended mRNA-Seq and Ribo-Seq data of 10 developmental stages and tissues of *Drosophila simulans* and paired-end mRNA-Seq data of 0-2 h, 2-6 h, 6-12 h, and 12-24 h *Drosophila melanogaster* embryos, were deposited in the China National Genomics Data Center Genome Sequence Archive (GSA) under accession

numbers CRA003198, CRA007425, and CRA007426. The mRNA-Seq and Ribo-Seq data for the different developmental stages and tissues of *Drosophila melanogaster* were published in our previous paper and were deposited in the Sequence Read Archive (SRA) under accession number SRP067542. All original code has been deposited on GitHub: https://github.com/lujlab/uORF_buffer (copy archived at *Lu, 2022*) and https://github.com/lujlab/Buffer_eLife2025 (copy archived at *Lu, 2025*).

The following datasets were generated:

| Author(s) | Year | Dataset title | Dataset URL | Database and Identifier |
|---|---|---|---|---|
| Sun Y, Duan Y, Gao P, Liu C, Jin K, Dou S, Tang W, Zhang H, Lu J | 2024 | Upstream open reading frames buffer translational variability during *Drosophila* evolution and development | https://ngdc.cncb.ac.cn/gsa/search?searchTerm=CRA003198 | China National Genomics Data Center Genome Sequence Archive, CRA003198 |
| Sun Y, Duan Y, Gao P, Liu C, Jin K, Dou S, Tang W, Zhang H, Lu J | 2024 | Upstream open reading frames buffer translational variability during *Drosophila* evolution and development | https://ngdc.cncb.ac.cn/gsa/search?searchTerm=CRA007425 | China National Genomics Data Center Genome Sequence Archive, CRA007425 |
| Sun Y, Duan Y, Gao P, Liu C, Jin K, Dou S, Tang W, Zhang H, Lu J | 2024 | Upstream open reading frames buffer translational variability during *Drosophila* evolution and development | https://ngdc.cncb.ac.cn/gsa/search?searchTerm=CRA007426 | China National Genomics Data Center Genome Sequence Archive, CRA007426 |

The following previously published dataset was used:

| Author(s) | Year | Dataset title | Dataset URL | Database and Identifier |
|---|---|---|---|---|
| Zhang H, Dou S, He F, Luo J, Wei L, Lu J | 2018 | Genome-wide maps of ribosomal occupancy provide insights into adaptive evolution and regulatory roles of uORFs during *Drosophila* development | https://www.ncbi.nlm.nih.gov/sra/?term=SRP067542 | NCBI Sequence Read Archive, SRP067542 |

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

## Appendix 1

### The detailed operations per action in the ICIER simulation

The translation process of an mRNA molecule is modeled as a series of discrete actions. During each action, certain events may occur with certain probabilities for each entity (40S or 80S ribosome). The operations per action are described as follows:

If the first 10 triplets were vacant of the mRNA, a 40S ribosome would be added there with a probability of $R_{in}$.

2. If a 40S ribosome was located at the start codon of the uORF, it would transform into an 80S ribosome with a probability of $I_{uORF}$. If a 40S ribosome was located at the start codon of the CDS, it would transform into an 80S ribosome with a probability of $I_{CDS}$.

3.If an 80S ribosome was located at the stop codon of the uORF or CDS, it would dissociate from the mRNA. The dissociation of an 80S ribosome from the uORF or CDS stop codon was counted as a completed translation event, and the values of $N_{EC}$ and $N_{Eu}$ were incremented by one, respectively.

If a 40S ribosome arrived at the 3' end of the mRNA, it would dissociate from the mRNA.

5. For each 40S ribosome on the mRNA, if its next position was vacant (in the downstream-dissociation model) or not occupied by another 40S ribosome (in the upstream-dissociation model and double-dissociation model), it would move to the next position with a probability of $\nu_s$.

6. For each 80S ribosome in the uORF, if its next position was vacant (in the upstream-dissociation model) or not occupied by another 80S ribosome (in the downstream-dissociation model and double-dissociation model), it would move to the next position with a probability of $\nu_{Eu}$. Similarly, for each 80S ribosome in the CDS, if its next position was vacant (in the upstream-dissociation model) or not occupied by an 80S ribosome (in the downstream-dissociation model and double-dissociation model), it would move to the next position with a probability of $\nu_{EC}$.

It was checked whether any 80S ribosome collided with a 40S ribosome. If such a collision occurred, the 40S ribosome involved would dissociate from the mRNA with a probability of $K_{up}$ (upstream dissociation or double-dissociation model) or $K_{down}$ (downstream dissociation or double-dissociation model).

These above operations are executed repeatedly until the action count reaches the limit given by the user (1,000,000 in this study).

