## [Editor Report · eLife Assessment]

This study reveals the **important** role of upstream open reading frames (uORFs) in limiting the translational variability of downstream coding sequences. Through a combination of computational simulations, comparative analyses of translation efficiency across different developmental stages in two closely related *Drosophila* species, and manipulative, experimental validation of translation buffering by an uORF for a gene, the authors provide **convincing** evidence supporting their conclusions. This work will be of broad interest to molecular biologists and geneticists.

---

## [Referee Report · Reviewer #1 (Public review)]

Summary:

The authors set out to explore the role of upstream open reading frames (uORFs) in stabilizing protein levels during *Drosophila* development and evolution. By utilizing a modified ICIER model for ribosome translation simulations and conducting experimental validations in *Drosophila* species, the study investigates how uORFs buffer translational variability of downstream coding sequences. The findings reveal that uORFs significantly reduce translational variability, which contributes to gene expression stability across different biological contexts and evolutionary timeframes.

Strengths:

(1) The study introduces a sophisticated adaptation of the ICIER model, enabling detailed simulation of ribosomal traffic and its implications for translation efficiency.

(2) The integration of computational predictions with empirical data through knockout experiments and translatome analysis in *Drosophila* provides a compelling validation of the model's predictions.

(3) By demonstrating the evolutionary conservation of uORFs' buffering effects, the study provides insights that are likely applicable to a wide range of eukaryotes.

Weaknesses:

(1) Although the study is technically sound, it does not clearly articulate the mechanisms through which uORFs buffer translational variability. A clearer hypothesis detailing the potential molecular interactions or regulatory pathways by which uORFs influence translational stability would enhance the comprehension and impact of the findings.

(2) The study could be further improved by a discussion regarding the evolutionary selection of uORFs. Specifically, it would be beneficial to explore whether uORFs are favored evolutionarily primarily for their role in reducing translation efficiency or for their capability to stabilize translation variability. Such a discussion would provide deeper insights into the evolutionary dynamics and functional significance of uORFs in genetic regulation.

Comments on revisions:

The authors have adequately addressed my previous concerns.

---

## [Referee Report · Reviewer #2 (Public review)]

uORFs, short open reading frames located in the 5' UTR, are pervasive in genomes. However, their roles in maintaining protein abundance are not clear. In this study, the authors propose that uORFs act as "molecular dam", limiting the fluctuation of the translation of downstream coding sequences. First, they performed in silico simulations using an improved ICIER model, and demonstrated that uORF translation reduces CDS translational variability, with buffering capacity increasing in proportion to uORF efficiency, length, and number. Next, they analysed the translatome between two related *Drosophila* species, revealing that genes with uORFs exhibit smaller fluctuations in translation between the two species and across different developmental stages within the same species. Moreover, they identified that bicoid, a critical gene for *Drosophila* development, contains a uORF with substantial changes in translation efficiency. Deleting this uORF in *Drosophila melanogaster* significantly affected its gene expression, hatching rates, and survival under stress conditions. Lastly, by leveraging public Ribo-seq data, the authors showed that the buffering effect of uORFs is also evident between primates and within human populations. Collectively, the study significantly advances our understanding of how uORFs regulate the translation of downstream coding sequences at the genome-wide scale, as well as during development and evolution. It would be particularly interesting to explore whether similar buffering functions are conserved in other organisms, and whether their regulatory effects could be harnessed for practical applications, such as improving crop traits or benefiting human health.

Comments on revisions:

The authors have fully addressed all of my concerns, and the revisions have substantially improved the manuscript. I have no further comments.

---

## [Author Response]

The following is the authors’ response to the original reviews

**Public Reviews:**

**Reviewer #1 (Public review):**
Summary:The authors set out to explore the role of upstream open reading frames (uORFs) in stabilizing protein levels during *Drosophila* development and evolution. By utilizing a modified ICIER model for ribosome translation simulations and conducting experimental validations in *Drosophila* species, the study investigates how uORFs buffer translational variability of downstream coding sequences. The findings reveal that uORFs significantly reduce translational variability, which contributes to gene expression stability across different biological contexts and evolutionary timeframes.

We thank the reviewer for carefully reading our manuscript and providing thoughtful and constructive feedback. We believe the manuscript has been significantly improved by incorporating your suggestions. Please find our detailed responses and corresponding revisions below.

Strengths:(1) The study introduces a sophisticated adaptation of the ICIER model, enabling detailed simulation of ribosomal traffic and its implications for translation efficiency.(2) The integration of computational predictions with empirical data through knockout experiments and translatome analysis in *Drosophila* provides a compelling validation of the model's predictions.(3) By demonstrating the evolutionary conservation of uORFs' buffering effects, the study provides insights that are likely applicable to a wide range of eukaryotes.

We appreciate your positive feedback and thoughtful summary of the strengths of our study.

Weaknesses:(1) Although the study is technically sound, it does not clearly articulate the mechanisms through which uORFs buffer translational variability. A clearer hypothesis detailing the potential molecular interactions or regulatory pathways by which uORFs influence translational stability would enhance the comprehension and impact of the findings.

Thanks for your constructive comments. In the Discussion section of our previous submission (Original Lines 470-489), we proposed that uORFs function as “molecular dams” to smooth out fluctuations in ribosomal flow toward downstream CDS regions, primarily via mechanisms involving ribosome collision and dissociation. To further address your concern, we have expanded the Discussion and included a new model figure (Fig. 9) to more clearly articulate the potential biological and mechanistic basis by which translating 80S ribosomes may induce the dissociation of 40S ribosomes. The revised section (Lines 540–557) now reads:

“Ribosome slowdown or stalling on mRNA due to rare codons [56,96-98] or nascent blocking peptides [99-102] frequently triggers ribosome collisions genome-wide [103-105]. Such collisions, especially among elongating 80S ribosomes, often activate ribosome quality control (RQC) pathways that recognize collision interfaces on the 40S subunit, leading to ribosomal subunit dissociation and degradation [106-108]. In mammals, ZNF598 specifically identifies collided ribosomes to initiate ubiquitin-dependent protein and mRNA quality control pathways [109-113]. Analogously, yeast employs Hel2-mediated ubiquitination of uS10, initiating dissociation via the RQC-trigger complex (RQT) [114]. Furthermore, the human RQT (hRQT) complex recognizes ubiquitinated ribosomes and induces subunit dissociation similarly to yeast RQT [115]. However, transient ribosome collisions can evade RQC by promoting resumed elongation through mechanical force provided by trailing ribosomes, thereby mitigating stalling [116]. Beyond 80S collisions, evidence increasingly highlights a distinct collision type involving scanning 40S subunits or pre-initiation (43S) complexes. Recently, an initiation RQC pathway (iRQC) targeting the small ribosomal subunit (40S) has been described, particularly involving collisions between scanning 43S complexes or between stalled 43S and elongating 80S ribosomes (Figure 9B) [117,118]. During iRQC, E3 ubiquitin ligase RNF10 ubiquitinates uS3 and uS5 proteins, resulting in 40S degradation [118]. This mechanism aligns closely with our ICIER model, proposing collision-driven 43S dissociation in the 5' UTRs. Future studies exploring these mechanisms in greater detail will clarify how uORFs modulate translational regulation through buffering effects.”

(2) The study could be further improved by a discussion regarding the evolutionary selection of uORFs. Specifically, it would be beneficial to explore whether uORFs are favored evolutionarily primarily for their role in reducing translation efficiency or for their capability to stabilize translation variability. Such a discussion would provide deeper insights into the evolutionary dynamics and functional significance of uORFs in genetic regulation.

Thank you for this insightful suggestion. We agree that understanding whether uORFs are evolutionarily favored for their role in translational repression or for their capacity to buffer translational variability is a compelling and unresolved question. Our study suggests that translational buffering, rather than translational repression alone, can also drive evolutionary selection favoring uORFs, although it remains challenging to empirically disentangle these functions due to their inherent linkage. We have expanded the discussion in the revised manuscript to address this point in more detail (Lines 494-513), which is reproduced as follows:

“Previous studies have shown that a significant fraction of fixed uORFs in the populations of *D. melanogaster* and humans were driven by positive Darwinian selection 63,67, suggesting active maintenance through adaptive evolution rather than purely neutral or deleterious processes. While uORFs have traditionally been recognized for their capacity to attenuate translation of downstream CDSs, accumulating evidence now underscores their critical role in stabilizing gene expression under fluctuating cellular and environmental conditions [43,55,56]. Whether the favored evolutionary selection of uORFs acts primarily through their role in translational repression or translational buffering remains a compelling yet unresolved question, as these two functions are inherently linked. Indeed, highly conserved uORFs tend to be translated at higher levels, resulting not only in stronger inhibition of CDS translation [34,45,67] but also in a more pronounced buffering effect, as demonstrated in this study. This buffering capacity of uORFs potentially provides selective advantages by reducing fluctuations in protein synthesis, thus minimizing gene-expression noise and enhancing cellular homeostasis. This suggests that selection may favor uORFs that contribute to translational robustness, a hypothesis supported by findings in yeast and mammals showing that uORFs are significantly enriched in stressresponse genes and control the translation of certain master regulators of stress responses [41,42,94,95]. Our study suggests that translational buffering, rather than translational repression alone, can also drive evolutionary selection favoring uORFs, although it remains challenging to empirically disentangle these functions. Future comparative genomic analyses, coupled with experimental approaches such as ribosome profiling and functional mutagenesis, will be crucial in elucidating the precise evolutionary forces driving uORF conservation and adaptation.”

**Reviewer #2 (Public review):**
uORFs, short open reading frames located in the 5' UTR, are pervasive in genomes. However, their roles in maintaining protein abundance are not clear. In this study, the authors propose that uORFs act as "molecular dam", limiting the fluctuation of the translation of downstream coding sequences. First, they performed in silico simulations using an improved ICIER model, and demonstrated that uORF translation reduces CDS translational variability, with buffering capacity increasing in proportion to uORF efficiency, length, and number. Next, they analzed the translatome between two related *Drosophila* species, revealing that genes with uORFs exhibit smaller fluctuations in translation between the two species and across different developmental stages within the same specify. Moreover, they identified that bicoid, a critical gene for *Drosophila* development, contains a uORF with substantial changes in translation efficiency. Deleting this uORF in *Drosophila melanogaster* significantly affected its gene expression, hatching rates, and survival under stress condition. Lastly, by leveraging public Ribo-seq data, the authors showed that the buffering effect of uORFs is also evident between primates and within human populations. Collectively, the study advances our understanding of how uORFs regulate the translation of downstream coding sequences at the genome-wide scale, as well as during development and evolution.The conclusions of this paper are mostly well supported by data, but some definitions and data analysis need to be clarified and extended.

We thank the reviewer for the thoughtful and constructive review. Your summary accurately captures the key findings of our study. We have carefully addressed all your concerns in the revised manuscript, and we believe it has been significantly improved based on your valuable input.

(1) There are two definitions of translation efficiency (TE) in the manuscript: one refers to the number of 80S ribosomes that complete translation at the stop codon of a CDS within a given time interval, while the other is calculated based on Ribo-seq and mRNA-seq data (as described on Page 7, line 209). To avoid potential misunderstandings, please use distinct terms to differentiate these two definitions.

Thank you for highlighting this important point, and we apologize for the confusion. The two definitions of translation efficiency (TE) in our manuscript arise from methodological differences between simulation and experimental analyses. To clarify, in the revised manuscript, we use “translation rate” in the context of simulations to describe the number of 80S ribosomes completing translation at the CDS stop codon per unit time. We retain the conventional “translation efficiency (TE)” for Ribo-seq–based measurements.

In this revised manuscript, we have added a more detailed explanation of TE in the revised manuscript (Lines 202–206), which now reads:

“For each sample, we followed established procedures [62-66] to calculate the translational efficiency (TE) for each feature (CDS or uORF). TE serves as a proxy for the translation rate at which ribosomes translate mRNA into proteins, typically quantified by comparing the density of ribosome-protected mRNA fragment (RPF) to the mRNA abundance for that feature (see Materials and Methods).”

(2) Page 7, line 209: "The translational efficiencies (TEs) of the conserved uORFs were highly correlated between the two species across all developmental stages and tissues examined, with Spearman correlation coefficients ranging from 0.478 to 0.573 (Fig. 2A)." However, the authors did not analyze the correlation of translation efficiency of conserved CDSs between the two species, and compare this correlation to the correlation between the TEs of CDSs. These analyzes will further support the authors conclusion regarding the role of conserved uORFs in translation regulation.

In the revised manuscript, we have incorporated a comparison of translational efficiency (TE) correlations for conserved CDSs between the two species. We found that CDSs exhibit significantly higher interspecific TE correlations than uORFs, with Spearman’s *rho* ranging from 0.588 to 0.806. This suggests that uORFs tend to show greater variability in TE than CDSs, consistent with our model in which uORFs buffer fluctuations in downstream CDS translation. The updated results were included in the revised manuscript (Lines 223-227) as follows:

“In contrast, TE of CDSs exhibited a significantly higher correlation between the two species in the corresponding samples compared to that of uORFs, with Spearman’s *rho* ranging from 0.588 to 0.806 (*P* = 0.002, Wilcoxon signed-rank test; Figure 2A). This observation is consistent with our simulation results, which indicate that uORFs experience greater translational fluctuations than their downstream CDSs.”

(3) Page 8, line 217: "Among genes with multiple uORFs, one uORF generally emerged as dominant, displaying a higher TE than the others within the same gene (Fig. 2C)." The basis for determining dominance among uORFs is not explained and this lack of clarification undermines the interpretation of these findings.

Thank you for pointing this out. We apologize for the confusion. In our study, a “dominant” uORF is defined as the one with the highest translation efficiency (TE) among all uORFs within the same gene. This designation is based solely on TE, which we consider a key metric for uORF activity, as it directly reflects translational output and potential regulatory impact. We have revised the manuscript to clarify this definition (Lines 232–244), now stating:

“Among genes with multiple uORFs, we defined the uORF with the highest TE as the dominant uORF for that gene, as TE is one of the most relevant metrics for assessing uORF function 45,67…… These results suggest that genes with multiple uORFs tend to retain the same dominant uORF across developmental stages, indicating that the dominant uORFs may serve as the key translational regulator of the downstream CDS.*”*

(4) According to the simulation, the translation of uORFs should exhibit greater variability than that of CDSs. However, the authors observed significantly fewer uORFs with significant TE changes compared to CDSs. This discrepancy may be due to lower sequencing depth resulting in fewer reads mapped to uORFs. Therefore, the authors may compare this variability specifically among highly expressed genes.

Thank you for this thoughtful observation. We agree that the lower proportion of uORFs showing significant TE changes compared to CDSs, as reported in Table 1, appears inconsistent with our conclusion that uORFs exhibit greater translational variability. However, this discrepancy is largely attributable to differences in sequencing depth and feature length—uORFs are generally much shorter and more weakly expressed than CDSs, resulting in fewer mapped reads and reduced statistical power (Figure S18A).

To address this issue, we first followed your suggestion and restricted our analysis to genes with both mRNA and RPF RPKM values above the 50th percentile in *D. melanogaster* and *D. simulans*. While this filtering increased the total proportion of features with significant TE changes (due to improved read coverage), the proportion of significant uORFs still remained lower than that of CDSs (Table R1). This suggests that even among highly expressed genes, the disparity in read counts between uORFs and CDSs persists (Figure S18B), and thus the issue is not fully resolved.

To better capture biological relevance, we compared the absolute values of log2(TE changes) between *D. melanogaster* and *D. simulans* for uORFs and their corresponding CDSs. Across all samples, uORFs consistently exhibit larger TE shifts than their downstream CDSs, supporting our model that uORFs act as translational buffers (Figure 3B).

We have made relevant changes to report the new analysis in this revised manuscript. Specifically, in our original submission, we stated this observation with the sentence “The smaller number of uORFs showing significant TE changes compared to CDSs between *D. melanogaster* and *D. simulans* likely reflects their shorter length and reduced statistical power, rather than indicating that uORFs are less variable in translation than CDSs.” To make this point clearer, in the revised version (Lines 275-284), we rephrased this sentence which read as follows:

“Note that due to their shorter length and generally lower TE, uORFs had considerably lower read counts than CDSs, limiting the statistical power to detect significant interspecific TE differences for uORFs. This trend consistently holds whether analyzing all expressed uORFs (Figure S18A) or only highly expressed genes (Figure S18B). Thus, the fewer uORFs showing significant TE divergence likely reflects lower read counts and statistical sensitivity rather than reduced translational variability relative to CDSs. In fact, the absolute values of log2(fold change) of TE for uORFs between *D. melanogaster* and *D. simulans* were significantly greater than those observed for corresponding CDSs across all samples (*P* < 0.001, Wilcoxon signed-rank test; Figure 3B), suggesting that the magnitude of

TE changes in CDSs is generally smaller than that in uORFs, due to the buffering effect of uORF.”

**Author response table 1. sa3table1:** Proportion of uORFs and CDSs with significant TE changes before and after selecting HEGs.

	Original				Highly expressed genes			
Sample types	# of expressed uORFs	beta_("III ")!=1 (%)	# of expressed CDSs	b_(sigma)!=1 (%)	# of expressed uORFs	beta_("III ")!=1 (%)	# of expresse d CDSs	b_(c)!=1 (%)
0-2 h embryo	7,704	1,193 (15.49)	7,934	4,189 (52.80)	3,531	703 (19.90)	2760	1,393 (50.47)
2-6h embryo	7,822	567 (7.25)	9,795	3,063 (31.27)	4,666	456 (9.77)	3910	1,926 (49.26)
6-12h embryo	10,400	1,040 (10.00)	10,643	2,924 (27.47)	5,718	866 (15.15)	4271	1,544 (36.15)
12-24h embryo	11,365	535 (4.71)	11,158	3,537 (31.70)	5,454	423 (7.76)	4304	1,852 (43.03)
Larva	10,008	464 (4.64)	11,831	3,554 (30.04)	5,247	377 (7.18)	4546	2,149 (47.27)
Pupa	12,309	635 (5.16)	12,209	4,087 (33.48)	6,028	480 (7.96)	4487	2,196 (48.94)
Male body	10,894	197 (1.81)	12,284	2,432 (19.80)	4,422	135 (3.05)	4461	1,390 (31.16)
Male head	10,904	144 (1.32)	10,447	1,151 (11.02)	5,644	123 (2.18)	3981	605 (15.20)
Female body	9,809	340 (3.47)	11,002	3,605 (32.77)	4,231	255 (6.03)	4064	2,047 (50.37)
Female head	10,935	332 (3.04)	10,545	1,270 (12.04)	5,510	269 (4.88)	4019	618 (15.38)

(5) If possible, the author may need to use antibodies against bicoid to test the effect of ATG deletion on bicoid expression, particularly under different developmental stages or growth conditions.According to the authors' conclusions, the deletion mutant should exhibit greater variability in bicoid protein abundance. This experiment could provide strong support for the proposed mechanisms.

Thank you for this excellent suggestion. We fully agree that testing Bcd protein levels across developmental stages or stress conditions using antibodies would be a strong validation of our model, which predicts greater variability in Bcd protein abundance upon uORF deletion.

In fact, we attempted such experiments in both wild-type and mutant backgrounds. However, we encountered substantial difficulties in obtaining a reliable anti-Bcd antibody. Some Bcd antibodies referenced in the published literature were homemade and often shared among research groups as gifts [1-3] and some commercially available antibodies cited in previous studies are no longer supplied by vendors [4-6]. We managed to obtain a custom-made antibody from Professor Feng Liu, but unfortunately, it produced inconsistent and unsatisfactory results. Despite considerable effort—including during the COVID-19 pandemic—we were unable to identify a reagent suitable for robust and reproducible detection of Bcd protein.

As an alternative, we used sucrose gradient fractionation followed by qPCR to directly measure the translation efficiency of *bicoid* in vivo. We believe this approach offers a clear and quantitative readout of translational activity, and it avoids potential confounding from protein degradation, which may vary across conditions and developmental stages. Nonetheless, we recognize the value of antibody-based validation and will pursue this direction in future work if reliable antibodies become available. We have added this limitation to the revised Discussion section (Lines 563–568) as follows:

“We demonstrated that the *bcd* uORF represses CDS translation using sucrose gradient fractionation followed by qPCR—an approach that directly measures translation efficiency while minimizing confounding from RNA/protein degradation. However, detecting Bcd protein levels with antibodies across developmental stages or conditions in the mutants and wild-type controls would provide an even stronger validation of our model and should be explored in future studies.”

**Recommendations for the authors:**

**Reviewer #1 (Recommendations for the authors):**
(1) The authors should provide a more detailed explanation for the modifications made to the ICIER model. Specifically, an explanation of the biological or mechanistic rationale behind the ability of the 80S ribosome to cause upstream 40S ribosomes to dissociate from mRNA would help clarify this aspect of the model.

Thank you for this suggestion. In the original submission, we described our modifications to the ICIER model in the section titled “An extended ICIER model for quantifying uORF buffering in CDS translation” (Lines 88-124 of the revised manuscript).

To further clarify the biological rationale behind this mechanism, we have now included a conceptual model figure (Figure 9) illustrating mechanistically how uORF translation can buffer downstream translation within a single mRNA molecule. Additionally, we expanded the Discussion to summarize the current understanding of how collisions between translating 80S ribosomes and scanning 40S subunits may lead to dissociation, referencing known initial ribosome quality control (iRQC) pathways. These revisions provide a clearer mechanistic framework for interpreting the buffering effects modeled in our simulations. The relevant part is reproduced from Discussion (Lines 540-557) which reads as follows:

“Ribosome slowdown or stalling on mRNA due to rare codons [56,96-98] or nascent blocking peptides [99-102] frequently triggers ribosome collisions genome-wide [103-105]. Such collisions, especially among elongating 80S ribosomes, often activate ribosome quality control (RQC) pathways that recognize collision interfaces on the 40S subunit, leading to ribosomal subunit dissociation and degradation [106-108]. In mammals, ZNF598 specifically identifies collided ribosomes to initiate ubiquitin-dependent protein and mRNA quality control pathways [109-113]. Analogously, yeast employs Hel2-mediated ubiquitination of uS10, initiating dissociation via the RQC-trigger complex (RQT) [114]. Furthermore, the human RQT (hRQT) complex recognizes ubiquitinated ribosomes and induces subunit dissociation similarly to yeast RQT [115]. However, transient ribosome collisions can evade RQC by promoting resumed elongation through mechanical force provided by trailing ribosomes, thereby mitigating stalling [116]. Beyond 80S collisions, evidence increasingly highlights a distinct collision type involving scanning 40S subunits or pre-initiation (43S) complexes. Recently, an initiation RQC pathway (iRQC) targeting the small ribosomal subunit (40S) has been described, particularly involving collisions between scanning 43S complexes or between stalled 43S and elongating 80S ribosomes (Figure 9B) [117,118]. During iRQC, E3 ubiquitin ligase RNF10 ubiquitinates uS3 and uS5 proteins, resulting in 40S degradation [118]. This mechanism aligns closely with our ICIER model, proposing collision-driven 43S dissociation in the 5' UTRs. Future studies exploring these mechanisms in greater detail will clarify how uORFs modulate translational regulation through buffering effects.”

(2) The figure legend references Figure 5C; however, this figure appears to be missing from the document.

We apologize for the oversight. The missing panel previously referred to as Figure 5C has now been incorporated into the revised Figure 6A. The figure and its corresponding legend have been corrected accordingly in the updated manuscript.

**Reviewer #2 (Recommendations for the authors):**
This is an important study that enhances our understanding of the roles of uORFs in translational regulation. In addition to the suggestions provided in the public review, the following minor points should be addressed before publication in eLife:(1) Page 7, line 207: "We identified 18,412 canonical uORFs shared between the two species (referred to as conserved uORFs hereafter)." The term "canonical uORFs" requires clarification. Does this refer to uORFs with specific sequence features, conservation, or another defining characteristic?

Thank you for pointing this out. We apologize for the lack of clarity. In our study, a canonical uORF is defined as an open reading frame (ORF) that initiates with a canonical AUG start codon located in the 5′ untranslated region (UTR) and terminates with a stop codon (UAA, UAG, or UGA) within the same mRNA. Conservation of uORFs is defined solely based on the presence of AUG start codons at orthologous positions in the 5′ UTR across species, regardless of differences in the stop codon.

To clarify this definition, we have revised the sentence as follows (Lines 213-219): “We focused on canonical uORFs that initiate with an ATG start codon in the 5′ UTR and terminate with a stop codon (TAA, TAG, or TGA). Because the ATG start codon is the defining feature of a canonical uORF and tends to be more conserved than its downstream sequence [67], we defined uORF conservation based on the presence of the ATG start codon in the 5′ UTR of *D. melanogaster* and its orthologous positions in *D. simulans*, regardless of differences in the stop codon. Using this criterion, we identified 18,412 canonical uORFs with conserved start codons between the two species.”

(2) Page 8, line 227: "Furthermore, the dominant uORFs showed a higher proportion of conserved uATGs than the other translated uORFs." There appears to be a typographical error. Should "other uATGs" instead read "other uORFs"?

Thank you for pointing this out. As we addressed in response to your previous concern, in this study, we defined uORF conservation primarily based on the presence of their start codon (uATG) both in *D. melanogaster* and the orthologous sites of *D. simulans*, as the start codon is the defining feature of a uORF and tends to be more conserved than the remaining sequence, as demonstrated in our previous study [7]. We used the term “conserved uATGs” to reflect this definition and believe it accurately conveys the intended meaning in this context.

(3) Page 8, line 240: "uORFs exhibited a significant positive correlation with the TE of their downstream CDSs in all samples analyzed (P < 0.001, Spearman's correlation)." A Spearman's rho of 0.11 or 0.21 may not practically represent a "significant" positive correlation. Consider rephrasing this as "a positive correlation."

Thank you for the suggestion. We have revised the sentence in the manuscript to read (Lines 257-259): “uORFs exhibited a modest, yet statistically significant, positive correlation with the TE of their downstream CDSs across all samples analyzed (*P* < 0.001, Spearman’s correlation).”

(4) Page 9, line 269: The analysis of interspecific TE changes between uORFs and their corresponding CDSs is a crucial piece of evidence supporting the authors' conclusions. Presenting this analysis as part of the figures, rather than in "Table 1," would improve clarity and accessibility.

Thank you for this suggestion. In Table 1, we originally presented the number of uORFs and CDSs that showed significant differences in TE between *D. melanogaster* and *D. simulans* during various developmental stages. One key point we aimed to emphasize was that, although TE changes in uORFs and their downstream CDSs are positively correlated, there is a notable difference in the magnitude of these changes. To better convey this, we have summarized the core findings of Table 1 in graphical form.

In Figure 3B of the revised version, we compared the absolute values of interspecific TE changes between CDS and uORF, showing that CDSs consistently exhibit smaller shifts than their upstream uORFs. This result further supports the translational buffering effect of uORFs on downstream CDS expression. We have included the updated results in the revised manuscript (Lines 281-284) as follows:

“In fact, the absolute values of log2(fold change) of TE for uORFs between *D. melanogaster* and *D. simulans* was significantly greater than that observed for corresponding CDSs across all samples (*P* < 0.001, Wilcoxon signed-rank test; Figure 3B), suggesting that the magnitude of TE changes in CDSs is generally smaller than that in uORFs, due to the buffering effect of uORF.”

(5) Page 9, line 279: The phrase "dominantly translated" needs clarification. Does it refer to Figure 2C, where one uORF is dominantly translated within a gene, or does it mean that the uORF's translation is higher than that of its corresponding CDS?

We apologize for the obscurity. The phrase "dominantly translated" means one uORF with the highest TE compared to other uORFs within a gene. We have rephrased the relevant sentence in the revised version (Lines 299-304), which now reads:

“To investigate how the conservation level and translation patterns of uORFs influence their buffering capacity on CDS translation, we categorized genes expressed in each pair of samples into three classes:

Class I, genes with conserved uORFs that are dominantly translated (i.e., exhibiting the highest TE among all uORFs within the same gene) in both *Drosophila* species; Class II, genes with conserved uORFs that are translated in both species but not dominantly translated in at least one; and Class III, the remaining expressed genes.”

(6) The sequencing data and analysis code should be made publicly available before publication to ensure transparency and reproducibility.

Thank you for this suggestion. As described in the Data availability section, all deepsequencing data generated in this study, including single-ended mRNA-Seq and Ribo-Seq data of 10 developmental stages and tissues of *Drosophila simulans* and paired-end mRNA-Seq data of 0-2 h, 26 h, 6-12 h, and 12-24 h *Drosophila melanogaster* embryos, were deposited in the China National Genomics Data Center Genome Sequence Archive (GSA) under accession numbers CRA003198, CRA007425, and CRA007426. The mRNA-Seq and Ribo-Seq data for the different developmental stages and tissues of *Drosophila melanogaster* were published in our previous paper [8] and were deposited in the Sequence Read Archive (SRA) under accession number SRP067542.

All original code has been deposited on GitHub: https://github.com/lujlab/uORF_buffer; https://github.com/lujlab/Buffer_eLife2025.

Response reference

(1) Li, X.Y., MacArthur, S., Bourgon, R., Nix, D., Pollard, D.A., Iyer, V.N., Hechmer, A., Simirenko, L., Stapleton, M., Luengo Hendriks, C.L., et al. (2008). Transcription factors bind thousands of active and inactive regions in the *Drosophila* blastoderm. PLoS Biol 6, e27. 10.1371/journal.pbio.0060027.

(2) Horner, V.L., Czank, A., Jang, J.K., Singh, N., Williams, B.C., Puro, J., Kubli, E., Hanes, S.D., McKim, K.S., Wolfner, M.F., and Goldberg, M.L. (2006). The *Drosophila* calcipressin sarah is required for several aspects of egg activation. Curr Biol 16, 1441-1446. 10.1016/j.cub.2006.06.024.

(3) Lee, K.M., Linskens, A.M., and Doe, C.Q. (2022). Hunchback activates Bicoid in Pair1 neurons to regulate synapse number and locomotor circuit function. Curr Biol 32, 2430-2441 e2433. 10.1016/j.cub.2022.04.025.

(4) Wharton, T.H., Nomie, K.J., and Wharton, R.P. (2018). No significant regulation of bicoid mRNA by Pumilio or Nanos in the early *Drosophila* embryo. PLoS One 13, e0194865. 10.1371/journal.pone.0194865.

(5) Wang, J., Zhang, S., Lu, H., and Xu, H. (2022). Differential regulation of alternative promoters emerges from unified kinetics of enhancer-promoter interaction. Nat Commun 13, 2714. 10.1038/s41467-022-30315-6.

(6) Xu, H., Sepulveda, L.A., Figard, L., Sokac, A.M., and Golding, I. (2015). Combining protein and mRNA quantification to decipher transcriptional regulation. Nat Methods 12, 739-742. 10.1038/nmeth.3446.

(7) Zhang, H., Wang, Y., Wu, X., Tang, X., Wu, C., and Lu, J. (2021). Determinants of genomewide distribution and evolution of uORFs in eukaryotes. Nat Commun 12, 1076. 10.1038/s41467-021-21394-y.

(8) Zhang, H., Dou, S., He, F., Luo, J., Wei, L., and Lu, J. (2018). Genome-wide maps of ribosomal occupancy provide insights into adaptive evolution and regulatory roles of uORFs during *Drosophila* development. PLoS Biol 16, e2003903. 10.1371/journal.pbio.2003903.